# CollabABILITY Cards: Supporting Researchers and Educators to Co-Design Computer-Supported Collaborative Learning Activities for Deaf Children

Leandro Flórez-Aristizábal [1,*], César Alberto Collazos [2], Sandra Cano [3] and Andrés Solano [4]

1   Faculty of Engineering, Institución Universitaria Antonio José Camacho, Cali 760046, Colombia
2   IDIS Research Group, Universidad del Cauca, Popayán 190003, Colombia
3   School of Informatics Engineering, Pontificia Universidad Católica de Valparaíso, Valparaíso 2340000, Chile
4   GITI Research Group, Universidad Autónoma de Occidente, Cali 760030, Colombia
*   Correspondence: learistizabal@admon.uniajc.edu.co; Tel.: +57-3002069670

**Abstract:** Implementing Collaborative Learning (CL) activities to support the education of children is something that must be carefully designed to achieve the desired goals since just having a group of children working on the same activity does not guarantee proper collaboration. It is something that goes from defining the profile of the students to structuring the collaboration according to the learning objectives, the number of children per group, roles defined, and shared resources among others. Designing Computer-Supported Collaborative Learning (CSCL) activities may be even harder to achieve when collaboration is trying to be accomplished by children with some kind of disability due to differences in the way they communicate or understand the world around them, which is why in this study we decided to focus on designing CSCL activities for deaf children. Since there is not a clear path in the literature to achieve effective collaboration among deaf learners, we propose four stages to be followed through a set of 27 cards that were designed to guide designers/developers and educators through the process of co-designing such activities. The cards were implemented in such a way that they were easy to follow along with, with templates that allowed designers of the CL activity to register all the information related to it. Digital and printed versions of the cards were evaluated by researchers and educators with satisfactory results and a prototype for mobile devices was developed and tested by children through individual and collaborative learning activities.

**Keywords:** Computer-Supported Collaborative Learning; deaf children education; design; user experience

## 1. Introduction

Collaborative Learning (CL) is an umbrella term for a variety of educational approaches the main purpose of which is to allow students to join intellectual efforts by working in groups and taking advantage of one another's skills or knowledge, mutually searching for understanding or solutions [1–3]. This approach has been shown to benefit learners at many levels (social, academic, and psychological) [4] as long as it is applied in carefully designed environments for this purpose [2]. Nowadays, technological advances that make it possible for students to interact with peers despite the distance have led to a research field known as Computer-Supported Collaborative Learning (CSCL) where technology supports collaboration among learners [5]. The inclusion of ICT makes CL processes more effective as it facilitates students' work and gives them independence; CL activities also become more engaging, especially for children [4–6].

Unfortunately, designing these types of activities to promote CL among children is not as easy as just grouping learners to work on a common task; CL must be achieved not only by understanding the students, but also the situations where and how the collaboration will take place. It is also important to notice that a collaborative activity needs to include at

least three aspects: equal participation, individual accountability, and positive interdependences (PI), which are at the heart of CL scenarios where learners understand that their individual success is linked to the success of every other member of the group [7]. With all these elements, an effective CL activity can be structured.

This study is part of a larger research project that aims to support literacy teaching to deaf children through interactive computer-supported collaborative tools [7]. Designing educational tools aimed at deaf children is a challenging task, especially for designers and developers of such tools, who do not understand the special needs of these children; thus, these are usually not taken into account in the design process [8]. It is not only deaf children who should participate in the design of a tool to be used by them, but also their teachers, who understand not just their needs, but also their skills and strengths. In order to ease the communication between educators and designers and thus successfully design collaborative learning activities that can be integrated into a technological tool, we propose a set of cards and templates that can guide this process as part of the DesignABILITY (Designing for Different Abilities) framework proposed in [7].

The CollabABILITY cards are structured in such a way that makes it easier to structure a CL activity as long as they are followed in the right order (defining children's profiles, setting initial conditions, structuring collaboration, and defining positive interdependences). These cards should be used primarily by educators who know better the pedagogical aspects and different abilities of children. Designers should take part in this process since they must evaluate how the proposed CL activity may be implemented in a technological tool. The fourth set of cards (positive interdependences) is also proposed to help designers define the right game mechanics that match positive interdependences to be included in the tool and also learning mechanics defined by educators.

The cards were implemented in both, physical cards, and a digital app. They were used and evaluated by educators of deaf children using an adapted System Usability Scale (SUS) evaluation [9]. The prototype with a CL activity was also evaluated by deaf children through usability tests. The results show that the cards are not just usable but also helpful in designing CL activities for deaf children.

The following section shows a background about the DesignABILITY framework which is the starting point of this study and where the idea of designing the CollabABILITY cards was conceived. Section 3 shows related work about collaborative learning activities designed for deaf children. In Section 4, the CollabABILITY cards are presented as well as their usability evaluations. Section 5 shows the design of a CL activity and its implementation in a digital prototype. In Section 6, a case study was carried out to evaluate the usability of the prototype with deaf children. Section 7 shows the evaluation of the prototype by educators and finally, Section 8 presents the conclusions of this study.

## 2. Background

The DesignABILITY framework [7] is proposed as an alternative to help designers and developers in the design of interactive collaborative tools to support teaching to children with disabilities. This framework is divided into four stages (learning requirements, design for engaged learning, prototyping, and evaluation) and it has been adapted for literacy teaching to deaf children through storytelling. For this adaptation, the design for the engaged learning stage involves storytelling and collaborative learning as strategies to motivate children and make the learning process meaningful (Figure 1).

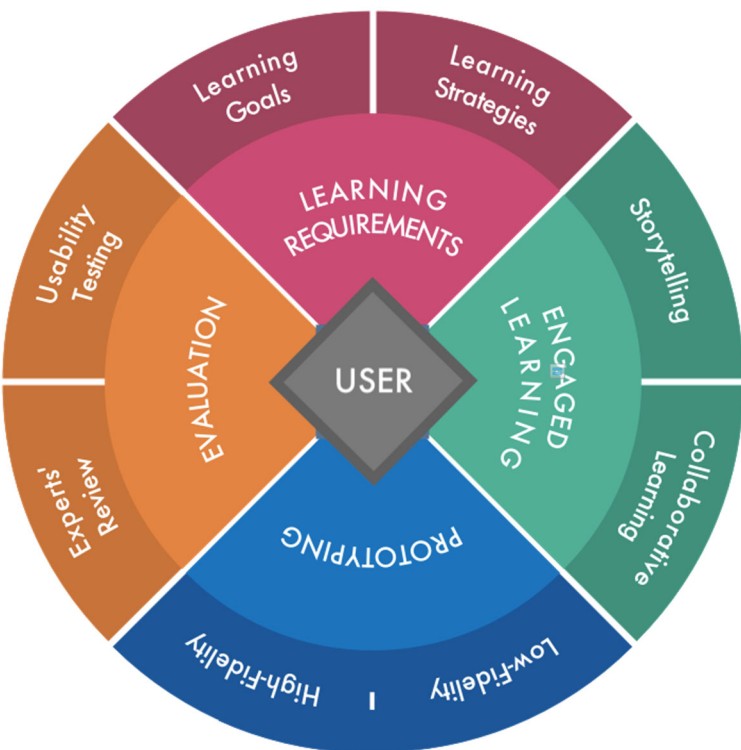

**Figure 1.** The DesignABILITY framework adapted for literacy teaching to deaf children [7].

The collaborative learning sub-stage defines the steps to be followed in the design of a CL activity. In the following section, the steps to be followed are explained.

### 2.1. Children's Profile

Here, non-sensitive data about the children must be gathered, such as age, gender and academic year, skills/abilities, learning methods and strategies, degree of hearing impairment, school level, use of sign language or cochlear implants, interests, and language level.

### 2.2. Initial Conditions

When the profile is defined, the situations where and how the collaboration will take place must be designed. This includes the type of activity, nature of collaborators, group heterogeneity, positive interdependences, setting of collaboration, conditions of collaboration, and period of collaboration.

### 2.3. Structure Collaboration

The collaboration is finally structured by defining these four elements: activities, roles, communication, and shared resources.

This sub-stage had been evaluated as a theoretical approach by expert reviewers in Human-Computer Interaction (HCI), designers, and software developers. Most of the reviewers agreed that even though it was well structured, they thought it could be hard to design a collaborative learning activity just by following this theoretical proposal. That is why the CollabABILITY cards and their templates were designed to make this process easier to execute.

While these three stages, along with positive interdependences, are necessary and well documented in the literature [5,6,9], implementing all these elements in one single design is not straightforward and there is no easy path to be followed, especially by people like designers or developers of technology who are not familiar with terms like collaborative learning or positive interdependences. Moreover, this process becomes more challenging when the people you design for are children with disabilities, for instance, deafness. For these reasons, the following research question arises: How can Computer-Supported

Collaborative Learning activities be designed for deaf children? To address this question, the following sections show related work that may contribute to this study, and a design approach is presented through a set of cards and templates.

## 3. Related Work

### 3.1. Collaborative Learning Design

Jeong and Hmelo-Silver proposed in [5] a set of seven affordances of CSCL that address particular types of functional requirements or challenges learners experience in collaborative settings. They can be further unpacked with a number of different dimensions with different technological solutions. The affordances proposed are:

- Collaborative tasks. Learners may want to collaborate, but they need a joint task that is group-worthy.
- Communication. For collaboration to occur, students need to be able to communicate with each other.
- Resources. Learners often possess unique experiences and expertise that are relevant to the task. One of the advantages of collaboration is that group members can combine the relevant resources distributed across the team.
- Structuring CL processes. Learners engage in different processes and activities during collaboration. They may ask and answer questions, criticize or provide feedback, and agree or disagree with other members of the group.
- Facilitating co-construction. Collaboration, by definition, means that partners work toward a shared goal and co-construct something new.
- Monitoring and regulation. In any kind of activity, metacognitive awareness and regulation are integral to good performance. Learners need to plan, monitor, and regulate CL processes.
- Forming and building groups and communities. Collaboration is a meeting of minds. For that to happen, students need to find partners or groups to work with and in which to work. Finding the right partners or groups who share the same interest as well as commitments to the task is key to the success of every collaboration.

These affordances are very valuable to this research and most of them are closely related to our proposal, but they do not involve any deaf-related information, they were proposed more as a general-purpose approach for CL.

Alsumait and Fasial [10] proposed a roadmap to define an interactive collaborative tool architecture to improve learning outcomes for Arab deaf students. The proposal is based on five pillars:

1. Active learning. This is concerned with the techniques and methods that involve a student in constructive learning rather than being passive and listening to a traditional lecture.
2. Student activity space. It is a virtual space with real-time capabilities where deaf members are offered a self-paced tool to acquire knowledge, view different resources, become engaged with the content, form groups, and assign different roles.
3. Technology used. E-learning technology is mostly visual and very interactive, thus fitting the deaf students' learning style perfectly.
4. Communication. Deaf students must be urged to discuss and express their knowledge and acquired experience with their colleagues.
5. Assessment. Designing assessment activities can help deaf students to become lifelong learners and lead to effective interferences.

Unlike the affordances proposed in the previous study, this architecture does include deaf-related information to take into account when designing CL scenarios, which is very helpful for the purposes of our study.

Educational tools such as serious games must be conceived to be played with others. According to Jesse Schell in his book, "The Art of Game Design: A Book of Lenses," playing with other people is not just natural but the preferred way for people to play. Among the five reasons he gives, collaboration is listed because it allows one to partake in game actions

and employ game strategies that are impossible with just one person, and collaboration allows players to enjoy the deep pleasures that come from group problem-solving and being part of a successful team. One hundred lenses are given in this book for game design and many of them are related to positive interdependences that assure true collaboration; this work may be very useful for the purposes of this study since game mechanics and game design play an important role when designing tools for children. However, no accessibility information is considered as part of these lenses, so those that may be included in this study must be adapted to be considered when designing collaborative activities or games for children with disabilities [11].

### 3.2. Collaborative Learning Tools for the Deaf

Egusa et al. [12,13] developed a collaborative/interactive learning game for a puppet show system for deaf children where two different activities were proposed, one called "Jumping" (electricity generation game) and the other called "Filling in the Blanks". The former works by first selecting two players who stand in front of a Kinect sensor and must jump at the same time to create the electricity needed by the character of a story in the puppet show. The two players can consult each other in order to jump at the same time. Once they have jumped simultaneously for a predefined number of times, another pair of players takes over. This function encourages collaboration among players to play the game. In the second activity, the players are asked to teach the character of the story how to properly use grammar. The players are shown a sentence without a Japanese positional particle, and options of particles to fill in the blank. The results of this study show that using bodily movements encourages collaborative learning among deaf children and improves their motivation to learn grammar. Unfortunately, the study does not show how the collaborative learning activities were designed and how CL was structured.

Namatame and Matsuda [9] developed a peer review application for art in special education for hard-of-hearing children which consists of a basic evaluation function and a direct comment function that has been implemented in a tablet PC. Through the study, researchers explored the following issues in educational and digital entertainment technology: (1) applying the peer-review learning strategy to art education and (2) establishing a collaborative learning environment for art education. It enables students to back-review the comments they received from their peer reviews, and by doing this, students also learn how to write better reviews. This kind of meta-cognitive skill would also help students learn domain knowledge and skills. They found that hard-of-hearing students enjoyed collaborative learning using the application, but just as in the previous study, this one does not give any insights about how CL was thought or structured in the design of the application.

## 4. The CollabABILITY Cards

### 4.1. Version 1.0

The CollabABILITY Cards were created to support designers/developers, and educators in the design process of collaborative learning activities and tools. It is important to highlight that these are meant to be used only by them and not learners. When the theoretical part of this approach was first published, an expert review was carried out through a survey to evaluate the DesignABILITY framework [7] where one of the stages is this CL design proposal. A total of 26 researchers answered the questions of the survey, where 92.3% of them have experience with HCI, 46.2% with design, and 73.1% with software development. No deaf education educators were part of this first evaluation since it was meant to be made by the people who may use the whole DesignABILITY framework (not just the CL stage of the framework) in the design of educational tools.

From a 5-Likert scale rating (where 1 is very difficult and 5 is very easy), 16 researchers (61.5%) rated this CL design approach with a score of 3, while 8 researchers (30.8%) rated it with 4, and the remaining (7.7%) rated it with 5. The average evaluation is 3.54 which means that it is not easy to design CL activities with this proposal. Reviewers had the chance to comment on their evaluation and most of them agreed that due to the extent

of the information provided for the design of CL activities, it was not easy to understand and implement all the steps suggested. Based on these results, we decided to design the CollabABILITY cards to make this process easier to follow.

Based on the theoretical approach presented in [7], there were 43 cards in the first version of the cards divided into four categories, and a different color was assigned to each category. These were tagged with letters A, B, C, and D and ascending numbers from 1 to N, where N is the last card of each category. Additionally, some cards that are complementary to others have a sub-tag _a, _b, _c, _d:

1. Children's Profile (A, A1, A2, A3)
2. Initial Conditions (B, B1, B2, B3, B4 -> D, B5, B6, B7)
3. Structure Collaboration (C, C1, C1_a, C2, C2_a, C3, C4, C4_a)
4. Positive Interdependences (D, D1, D1_a, D2, D3, D3_a, D3_b, D3_c, D4, D4_a, D5, D5_a, D5_b, D6, D6_a, D7, D7_a, D8, D8_a, D9, D9_a, D9_b, D9_c)

Figure 2 shows all the cards with the route to be followed.

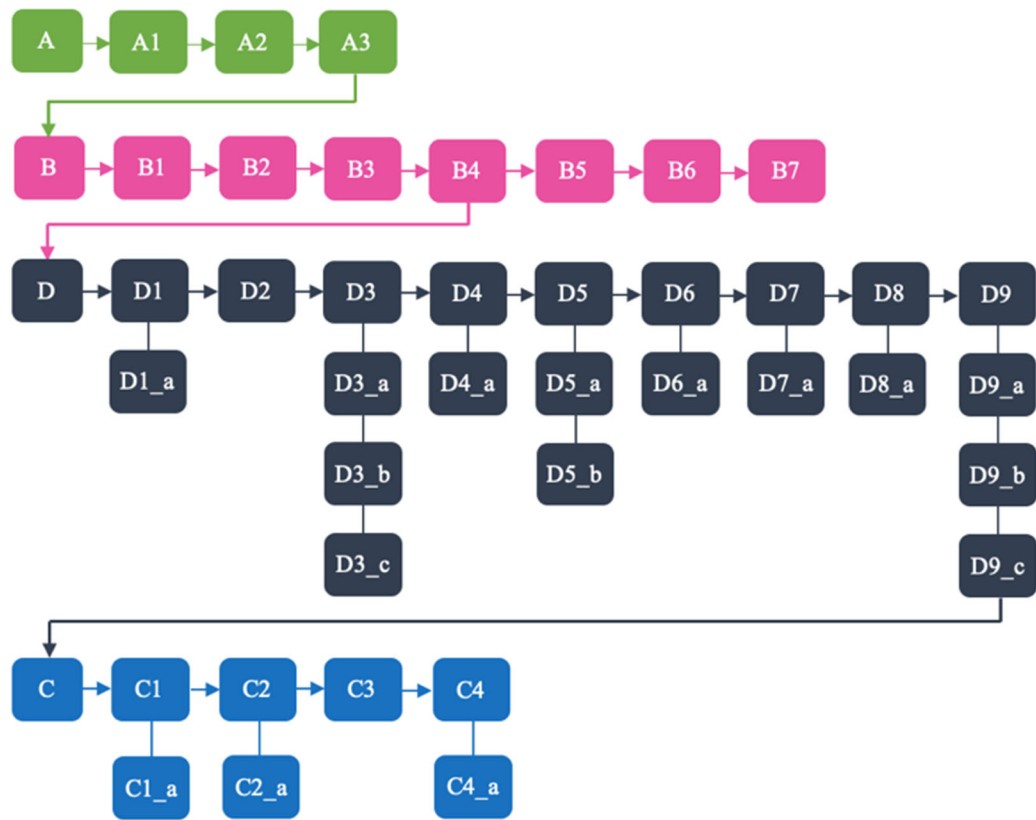

**Figure 2.** Path used in the first version of the cards.

Templates were created to be used along with the cards, so the information needed during the design of the CL activity is gathered and that way has a record of every step of the design process. Only one set of cards and templates is needed per working group (designer/developer, educator).

### 4.1.1. Evaluation of CollabABILITY Cards (Version 1.0)

Once the first version of the cards was designed, another evaluation was conducted with four experts, two of them researchers (one man and one woman between 30 and 40 years old) with an HCI/software development background from Cali (Colombia) and two educators, one of them is a school teacher of deaf children in Gourock (Scotland) and the other one is a school teacher of hearing children in Cali (Colombia), both of them women

between 30 and 40 years old. They worked in pairs (one researcher and one educator) and they were asked to use the cards in the design of a collaborative learning activity.

This version of the cards was evaluated along with their respective templates of every category through an adapted System Usability Scale (SUS) and a short questionnaire. The reason to use an adapted SUS was that these cards would be later implemented as a mobile app.

The 10 SUS statements presented to the experts were:

1.  I think that I would like to use these cards/templates frequently.
2.  I found the cards/templates unnecessarily complex.
3.  I thought the cards/templates were easy to use.
4.  I think that I would need the support of a technical person to be able to use these cards/templates.
5.  I found the various functions in these cards/templates were well integrated.
6.  I thought there was too much inconsistency in these cards/templates.
7.  I would imagine that most people would learn to use these cards/templates very quickly.
8.  I found the cards/templates very cumbersome to use.
9.  I felt very confident using the cards/templates.
10. I needed to learn a lot of things before I could get going with these cards/templates.

The response format used for each statement can be seen in Figure 3.

| Strongly Disagree 1 | 2 | 3 | 4 | Strongly Agree 5 |
|---|---|---|---|---|
| ☐ | ☐ | ☐ | ☐ | ☐ |

**Figure 3.** Response format for the SUS evaluation.

A series of open-ended questions were asked after the SUS evaluation to give the evaluator the opportunity to express what s/he thinks about the cards, the process, the time invested during the design of the collaborative learning activity, and how these could be improved.

### 4.1.2. Results

The 10 statements of the SUS evaluation were presented to both researchers (R1, R2) and educators (E1, E2) and their responses are shown in Figure 4. The SUS score was mapped along with adjective and acceptability ranges proposed by Bangor et al. in [14]

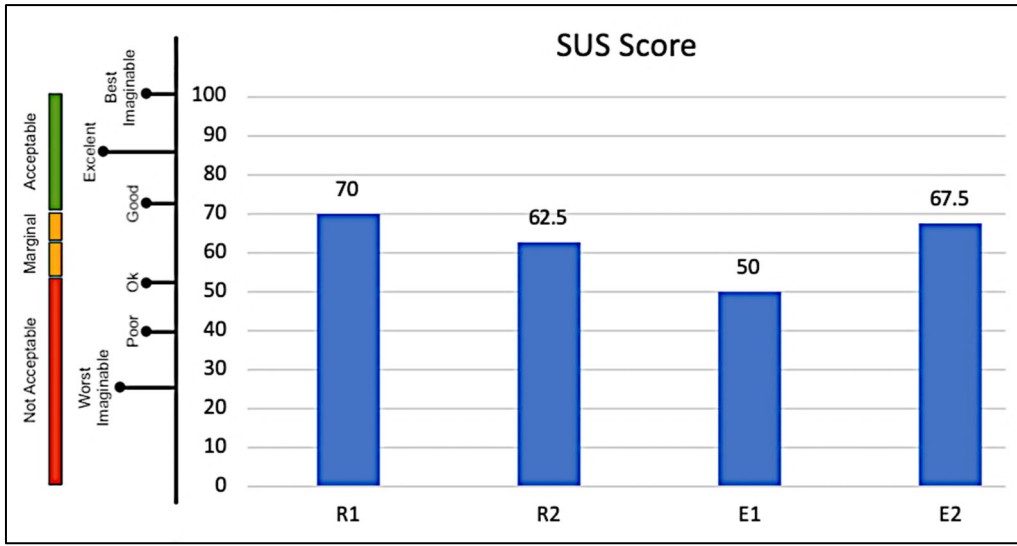

**Figure 4.** SUS Score mapped with adjective and acceptability ratings for version 1.0.

### 4.1.3. Discussion

After analyzing the SUS scores obtained, it is clear that the cards were not entirely usable; they reflect a marginal acceptance of usability which is not good enough for a tool that is supposed to make the design of a collaborative learning activity easier.

The open-ended questions show that the cards are complex to use mainly because of the number of cards to use. On one hand, they all suggested simplifying the process by reducing the number of cards and perhaps using less jargon. They also recommended some kind of training or tutorial in order to explain how the cards must be used. On the other hand, they think that using the cards regularly would reduce the time invested in the design of an activity, that the whole concept of the cards is a great idea, and that the cards were actually helpful for them despite the complexity.

### 4.2. Version 2.0

The cards were re-designed, dividing them once again into four categories, each category defined by a color and a letter, and each card numbered from 1 to N, where N is the last card of each category. The cards and the templates follow a fixed path during the design of a collaborative learning activity. These cards are tagged with letters from A to D and numbers in ascending order (1, 2, 3, . . . N).

The number of cards was reduced from 43 to 27 (a reduction of 37%) by simplifying the content of each card and thus eliminating the need for complementary cards tagged with (_a, _b, _c, _d) and making the process more straightforward. The use of the templates was also simplified by reducing an element of repetition that was identified in different categories.

The new path for this version of the cards goes as follows (Figure 5):

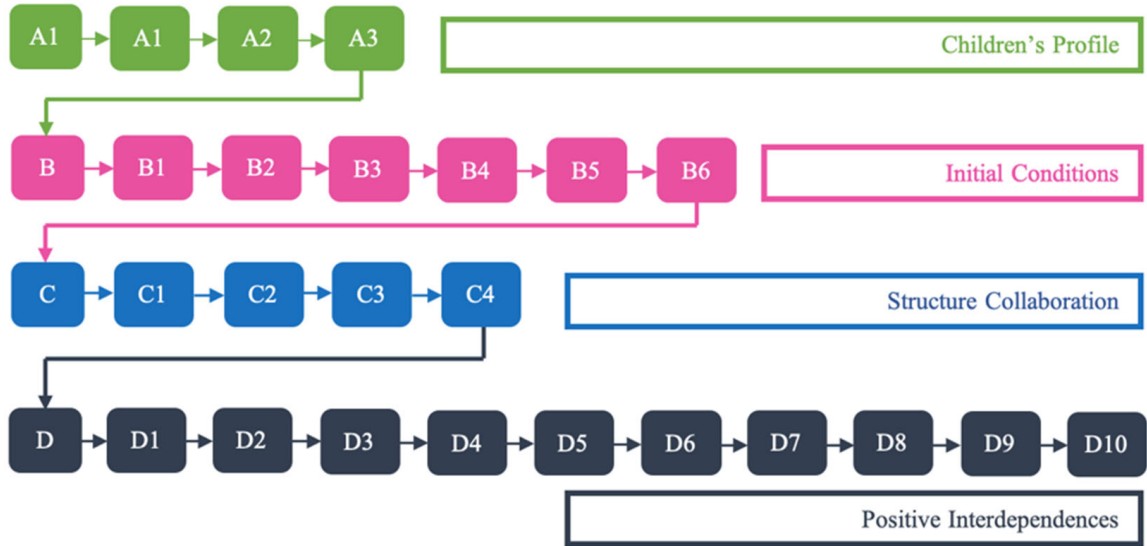

**Figure 5.** Path used in the final version of the cards.

The front side of the card shows the title of the subcategory and the back side shows additional information about what has to be done during the design.

The first category is *Children's profile*, tagged with the letter A, and it consists of four cards (A, A1, A2, A3). Figure 6 shows what the first category looks like (the first two cards). The front side of the card is on the left, and the back side is on the right and Table 1 shows all the details.

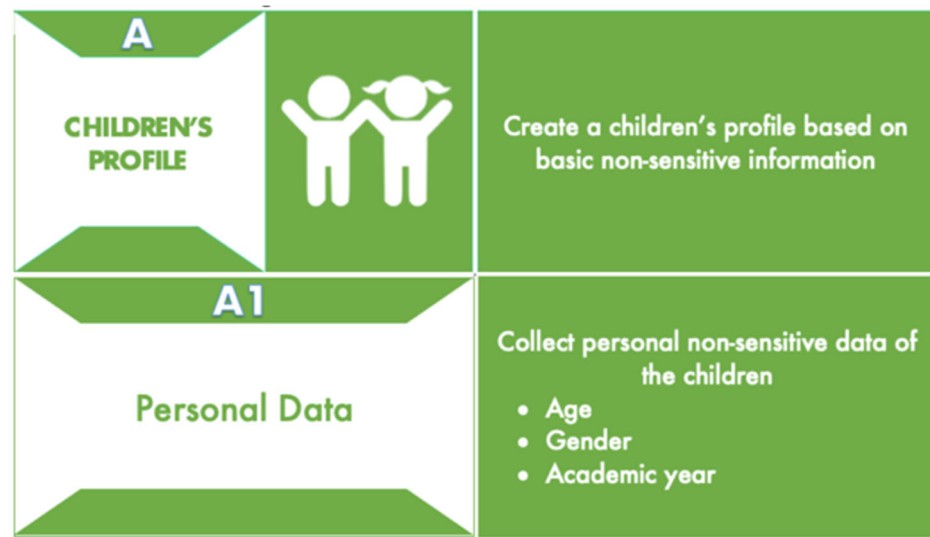

**Figure 6.** Children's profile category.

**Table 1.** Children's profile cards.

| Card | Front | Back |
|------|-------|------|
| A | This is the front card of this category | Explains that a children's profile must be created based on non-sensitive information |
| A1 | Personal Data | Shows the information to be collected (Age, gender, academic year) |
| A2 | Deafness-related information | Information such as degree of hearing loss, use of sign language, cochlear implants, etc. |
| A3 | Learning-related information | Data about children's skills, abilities, learning methods and strategies, school level, interests, language level |

The second category is Initial conditions tagged with letter B and it consists of seven cards (B, B1 to B6). Figure 7 shows what this category looks like (Cards B, B1) and Table 2 shows all the details.

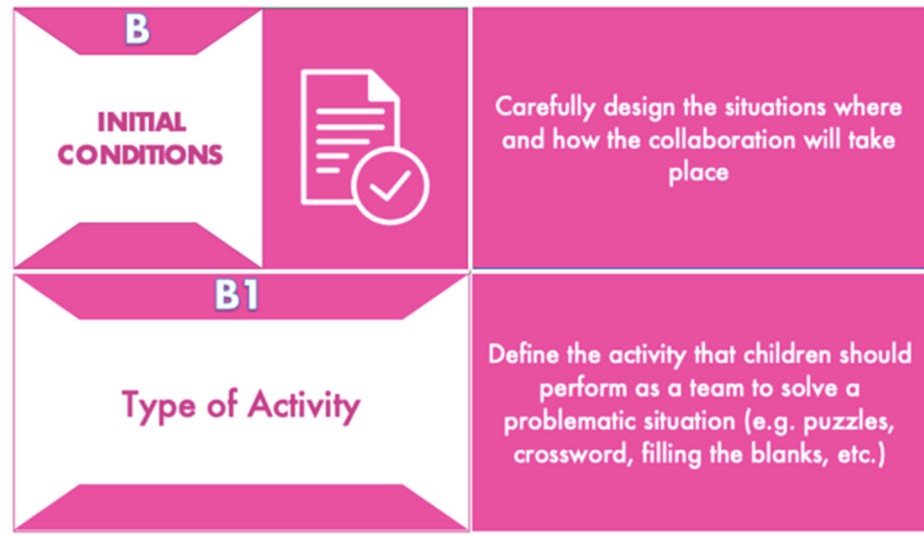

**Figure 7.** Initial conditions category.

**Table 2.** Initial conditions cards.

| Card | Front | Back |
|------|-------|------|
| B | This is the front card of this category | Explains that this category is about designing the situations where and how the collaboration will take place |
| B1 | Type of activity | Define the activity that children should perform as a team to solve a problematic situation (e.g., puzzles, crossword, filling the blanks, etc.). |
| B2 | Nature of collaborators | Specify the type of interaction (peer-to-peer, teacher-student, student-computer). |
| B3 | Group heterogeneity | Define variables such as size of the group, gender, or academic level. |
| B4 | Setting of collaboration | Define the place where the collaborative activity should take place (e.g., classroom, home, virtual environment). |
| B5 | Conditions of collaboration | Define how the collaboration will be mediated (physically, computer-mediated) and if it will be synchronous or asynchronous. |
| B6 | Period of collaboration | Time that will be invested by children during the activity. |

The third category is *Structure collaboration* tagged with the letter C and consists of five cards (C, C1 to C4). Figure 8 shows what this category looks like and Table 3 shows the details.

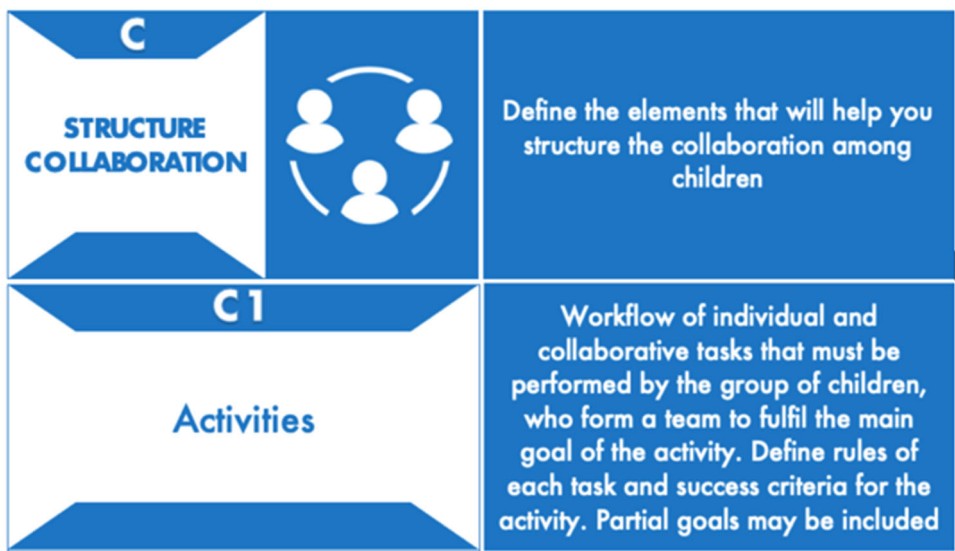

**Figure 8.** Structure collaboration category.

Finally, the last category was created for positive interdependences (PI). As mentioned before, PI is necessary to assure collaboration between members of a team. Johnson and Johnson [15–17] states that team members perceive that they need each other in order to complete the group's task ("sink or swim together"). To successfully design a CL activity, 9 positive interdependences are defined and one or more can be integrated to achieve collaborative work. As part of this research, these PI were mapped [18] with learning mechanics and game mechanics from the LM-GM (Learning Mechanics–Game Mechanics) framework [19] as well as with collaborative game mechanics (CGM) from [20].

GM and CGM are the rules and procedures that provide interaction with a game and for CGM; these rules promote collaboration among players. LM are pedagogical practices that support learning [19].

**Table 3.** Structure collaboration cards.

| Card | Front | Back |
|------|-------|------|
| C | Front card of this category | Explains that this category is about defining the elements that will help to structure the collaboration among children |
| C1 | Activities | Workflow of individual and collaborative tasks that must be performed by the group of children, who form a team to fulfil the main goal of the activity. Define rules of each task and success criteria for the activity. Partial goals may be included. |
| C2 | Roles | Each member of the group should be assigned a role during the activity with its own responsibilities. The role of the teacher must be also defined. Every member should have the opportunity to play a different role to balance the workload of the activity. |
| C3 | Communication | During the activity, members of the group should have the means to communicate and coordinate properly among themselves (either by text or sign language). |
| C4 | Shared resources | Each member of the group should be provided with the necessary resources to achieve the partial and main goals. Resources will be shared and represent the knowledge each member has to contribute to the activity and the success of the group. |

From the LM-GM framework [19], we mapped its mechanics with the CGM given in [20]. Then, from collaborative learning literature, we mapped the positive interdependences found in collaborative situations [21] with the previous LM-GM-CGM mapping. This new mapping will allow us to determine how collaborative learning could be implemented during the design of a system along with GM and LM to increase motivation towards achieving group goals.

This last category is a great way to communicate ideas between educators, designers and developers. For instance, if an educator suggests that the activity should provide a learning mechanic such as *incentive*, this can be translated to a game designer language as a game mechanic such as *reward/penalty* which is present in all games. This can also be mapped as a positive interdependence (*celebration/reward*), which guarantees that the activity promotes some kind of collaboration.

This category is called Positive interdependences tagged with the letter D and consists of the following cards (D, D1 to D10). Figure 9 shows how this category looks like and Table 4 shows the details.

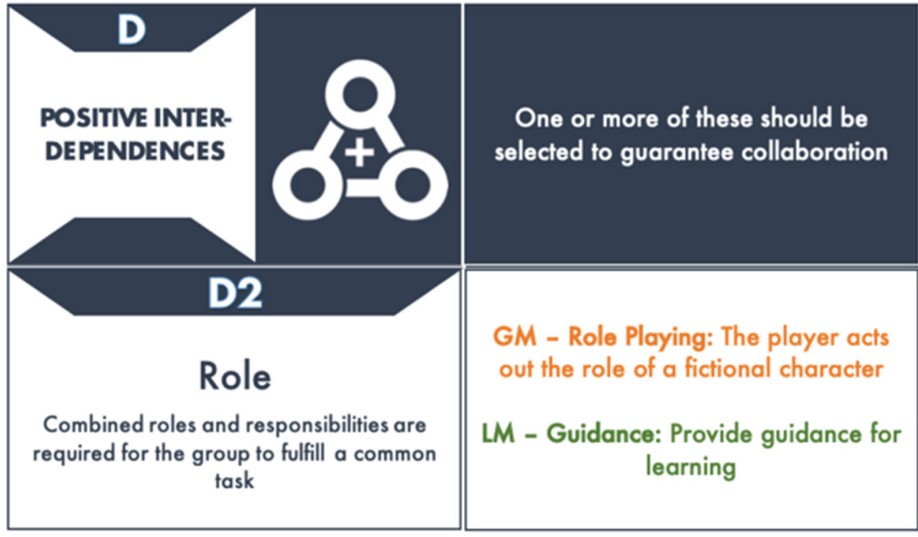

**Figure 9.** Positive Interdependences category.

**Table 4.** Positive interdependences cards.

| Card | Front (Positive Interdependence) | Back (Game Mechanics and Learning Mechanics) |
|------|----------------------------------|----------------------------------------------|
| D | This is the front card of this category | Explains that one or more PI should be included to guarantee collaboration. |
| D1 | Positive Interdependences (PI)<br>Game Mechanics (GM)<br>Learning Mechanics (LM) | Specify the types of PI that will assure true collaboration among students and encourage them to think as "we" instead of "me". GM (if necessary) and LM should also be specified in order to promote engagement and motivation in the learning activities |
| D2 | **PI–Role.** Combined roles and responsibilities are required for the group to fulfill a common task. | **GM–Role Playing:** The player acts out the role of a fictional character.<br>**LM–Guidance:** Provide guidance for learning. |
| D3 | **PI–Identity.** Makes unity and cohesion, increasing friendship and affinity through a shared identity expressed upon a common logo, motto, name, flag, or song. | No GM or LM were found for this PI. |
| D4 | **PI–Goal.** It is the belief that each team member can reach his or her goals only when the goals of the group are met. | **GM–Progression:** The success is granularly displayed and measured through the process of completing itemized tasks. Example: a progress bar.<br>**GM–Goal:** Sort of victory condition. Can be broad enough to encompass any method of winning, but here refers to game-specific goals. Example: Checkmate of a king in chess.<br>**GM–Cooperative play:** Encourages players to work together to beat the game. There is little or no competition between players. Either the players win the game, or all players lose it.<br>**LM–Collaborative:** More than one learner participates in a common learning activity to pursue a common goal.<br>**LM–Self–regulate:** Focus attention on one's own progress and cannel this towards achieving a goal.<br>**LM–Assist:** Help, promote, or support an equal or companion. |
| D5 | **PI–Environmental.** A physical environment that unifies the members of a group in which they work. No GM were found for this PI. | **LM–Situate:** Position learning in the context in which it is to be applied<br>**LM–Discover:** Gain understanding and solve problems by exploring/interacting with and manipulating the environment |
| D6 | **PI–Resource.** Each individual has only a part of the information, resources, or materials needed for his/her task. Therefore, the resources should be combined in order to accomplish the shared goal. | **GM–Communal discovery:** An entire community is rallied to work together to solve a problem/challenge. Immensely viral, a lot of fun.<br>**GM–Cascading information:** Information should be released in the minimum possible snippets to gain the appropriate level of understanding at each point during a game narrative.<br>**GM–Resource management:** The games' rules determine how players can increase, spend, or exchange their resources (tokens, money, etc.). The skillful management of resources under such rules allows players to influence the outcome of the game.<br>**LM–Connect:** Build knowledge by connecting information. |
| D7 | **PI–Task.** The organizing of the group works in a sequential pattern. When the actions of one group member have been accomplished, the next team member can proceed with his/her responsibilities. | **GM–Turn:** Segment of the game set aside for certain actions to happen before moving on to the next turn, where the sequence of events can largely repeat.<br>**LM–Master:** Proceed step by step, completing learning of one aspect before tackling a more difficult/complex one. |

| Card | Front (Positive Interdependence) | Back (Game Mechanics and Learning Mechanics) |
|---|---|---|
| D8 | **PI–Outside enemy.** Putting groups in competition with each other. Group members feel interdependent as they do their best to win the competition. | **GM–Micro leader–boards:** The rankings of all individuals in a micro-set. Often great for distributed game dynamics where you want many micro-competitions or desire to induce loyalty. Example: Be the top scorers at Joe's bar this week and receive a free appetizer.No LM were found for this PI. |
| D9 | **PI–Fantasy.** Giving an imaginary task to the students that requires members to assume they are in a life-threatening situation and their collaboration is needed to survive. | **GM–Narrative:** Draws the players into a story within the game. Example: Zombie Run, uses narrative to make the players believe that zombies are after them.<br><br>No LM were found for this PI. |
| D10 | **PI–Celebration/reward.** A mutual reward is given for successful group work and members' efforts to achieve it. | **GM–Achievement:** Segment A virtual or physical representation of having accomplished something. Often viewed as rewards. Example: A badge, a level, a reward, points.<br>**GM–Fixed ratio reward schedule:** Provides rewards after a fixed number of actions. This creates cyclical nadirs of engagement. Example: kill 20 ships, move a level up, receive a badge, visit five locations.<br>**GM–Chain schedule:** Linking a reward to a series of contingencies. Example: Kill 10 orcs to be let into the dragon's cave, every 30 min. the dragon appears.<br>**LM–Amplify:** Provide learner with high output in return for little input.<br>**LM–Reward:** Recognize achievement tangibly. |

This new version of the cards is available in both physical (printed) and digital versions. The digital version is an Android app that can be downloaded from the Play Store.

### 4.2.1. Evaluation of CollabABILITY Cards (Version 2.0)

A new evaluation was conducted with six experts from Colombia, three of them are researchers with an HCI/software development background, two men and one woman between 30 and 50 years old, and three school teachers (educators) of deaf children in Cali (Colombia), all three women between 30 and 40 years old. They worked in pairs (one researcher and one educator) and they were asked to use the cards in the design of a collaborative learning activity. One group of experts used the digital version of the cards (mobile app).

This new version of the cards was also evaluated along with their respective templates of every category through the adapted System Usability Scale (SUS) and the same questionnaire with a series of open-ended questions.

### 4.2.2. Results

Once again, the SUS score was mapped along with adjective and acceptability ranges proposed by Bangor et al. in [14].

The open-ended questions show some really good comments and reviews about the cards, such as: "the underlying idea was excellent and really helped me to develop the activity" says E1; "The cards were very appealing visually. Clear font, bright colors, and pleasant to use" says R2; "I think the cards are very useful as a stimulus for the teacher to consider how collaboration can best be achieved in an activity. I think a workshop on this design process would be very useful for student teachers and for experienced teachers. I would definitely use the cards again" states E3.

As a result of the activity, three collaborative learning activities were designed. Two of the activities ended as a paper prototype for desktop computers and one of them was implemented as a mobile app that was actually tested in a case study with deaf children in two Colombian educational institutions. Only the last CL activity (implemented as a mobile app) will be discussed in this study in Section 5.

4.2.3. Discussion

As can be seen in Figure 10, the usability of the cards increased greatly to an acceptable rating. Both the digital and printed versions of the cards received a high score which means that previous suggestions made by experts in the first evaluation were addressed correctly.

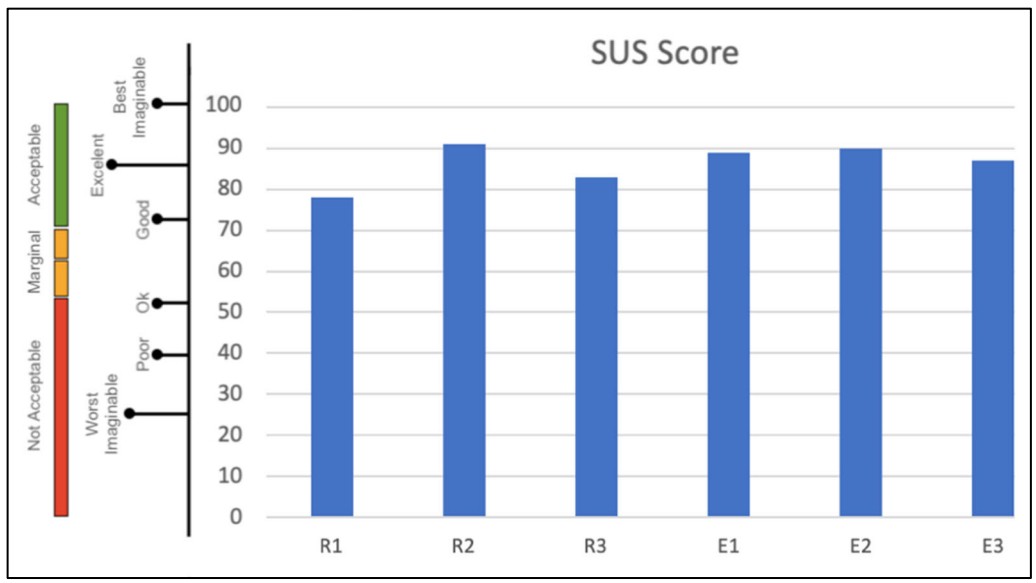

**Figure 10.** SUS Score mapped with adjective and acceptability ratings for version 2.0.

The qualitative evaluation made through the questionnaire reveals that the evaluators consider that this is actually a great resource for them, not just educators, but also researchers as it helps design collaborative learning activities that can be later integrated into a digital system.

The collaborative learning activity designed and implemented by one of the groups of experts demonstrates that co-designing this kind of activity requires an interdisciplinary team (designers and educators). All of the evaluators agreed that these kinds of resources (cards and templates) make this process easier as long as educators work along with designers as it is necessary to have a pedagogical and HCI/design/software development background.

## 5. Design of the Collaborative Learning Activity

*5.1. Materials and Methods*

- Participants

For the design of the collaborative learning activity, one educator and one software designer worked together to come up with an idea for a CL activity to be implemented in mobile devices. The cards were used as follows:

The educator was in charge of defining the children's profile taking into account that the activity is being designed for his/her students. To do so, the educator took each card of the children's profile category (Cards A) and after reading the information given by the card, the template (see Appendix B.2–Children's profile) was filled with the appropriate information.

- Children's Profile

  - Ages: 12–15
  - Gender: 4 girls and 2 boys
  - Academic year: 4 kids from 5th grade and 2 kids from 4th grade.
  - All 6 children are profoundly deaf, 1 of them has Jervell and Lange-Nielsen syndrome and another has a cognitive deficit, probably due to a mental disorder
  - All 6 children use sign language to communicate, none of them have a cochlear implant

○    Their literacy skills are very low, they only know a few words of written Spanish, no grammar, no reading comprehension

Once the children's profile was defined, both the educator and the designer decided which would be the initial conditions for the activity. This included the type of activity, how the children would interact, how the groups would be formed, and where and how the collaboration would take place. By using the second set of cards (B), the educator and designer filled the corresponding template to define the initial conditions (see Appendix B.2–Initial conditions) after reading the information given by cards B to B6. The cards give the options to be considered for the activity and general information on what could be included by the educator and the designer, and it is up to them to decide what is better in terms of learning and development.

- Initial Conditions
  - ○ Type of activity: Charades-type activity
  - ○ Interaction: peer-to-peer
  - ○ Group heterogeneity: Mixed (boys and girls when possible) and mixed academic level
  - ○ Setting of collaboration: Classroom
  - ○ Condition of collaboration: Physically
  - ○ Period of collaboration: 10 min

With the initial conditions set, the collaboration was structured by defining the activities to be performed (tasks, workflow) as well as the roles and resources to be shared. This process was carried out by the educator and designer by following the third and fourth sets of cards (C and D) and the templates for them (see Appendix B.2–Structure collaboration and Positive interdependences).

- Structure Collaboration
  - ○ Activities: Students will have a short period to learn new vocabulary, then, they must take turns (changing roles), while one of them sees the word, to find and sign it to his/her peer (this is called THE SIGNER); the other one (this is called THE WORD FINDER) has four options on the tablet to choose from. The main goal is to choose the correct words on the tablet based on an image shown to both students, THE SIGNER may help the WORD FINDER just by signing the word to be found. Since they must take turns, partial and individual goals consist of choosing the right word in every turn and the resource they share will be through sign language.
  - ○ Roles: THE SIGNER will be the student who sees the word and makes the sign to his peer. THE WORD FINDER will be the student holding the tablet with four options to choose from and matching the sign given by his/her peer with one of the words on the screen. The teacher moderates the activity making sure they don't break the rules (no watching the word seen by THE SIGNER or helping THE WORD FINDER to select the right word).
  - ○ Communication: Members of the team communicate through sign language.
  - ○ Shared resources: Each student will have a resource to complete the activity. On one hand, THE SIGNER has the word to be found and THE WORD FINDER will have the options to choose from.

- Positive Interdependences
  - ○ Role: Two roles must be played, taking turns.
  - ○ Identity: The students must come to terms and choose a flag to identify them as a team.
  - ○ Goal: Partial goals and one main goal (Selecting the correct words based on the sign).
  - ○ Resource: Two resources are shared.
  - ○ Task: They take turns every time one word is correctly matched with the sign.
  - ○ Celebration/Reward: Every time a word is correctly matched, the students will receive a STAR and a numeric score.

### 5.2. Prototype Development

A mobile prototype was developed, focusing on the learning goals and strategies defined by the teachers. After designing the activity with the CollabABILITY cards, a low-fidelity prototype was designed.

On the left, Figure 11 shows the screen where the children will see the story to be told and where the new vocabulary will come from.

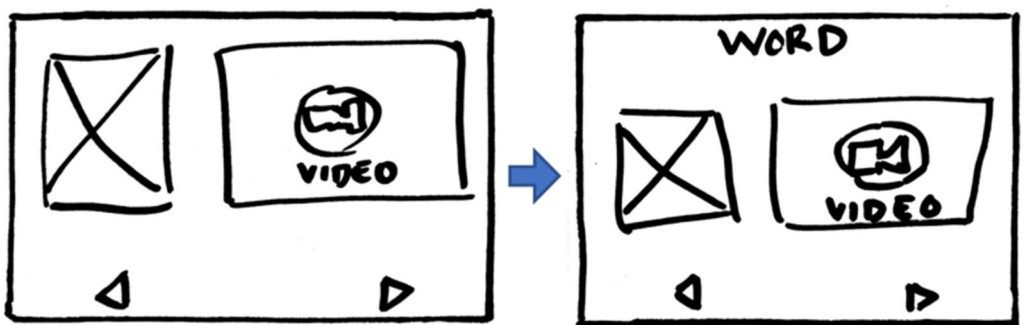

**Figure 11.** Storytelling screen and Word Training screen.

The content of the screen (left) is:

- An image with the scene of the story.
- A video with an interpreter telling the scene in sign language.
- Two buttons to go to the previous/next scene.

On the right is the screen where children will train to learn the new vocabulary from the story.

- The content of this screen (right) is:
- The word to be learned on top.
- An image that represents that word.
- A video with the sign of the word.
- The buttons to go to the previous/next word to learn.

Finally, on the left, Figure 12 shows the screen that will see the WORD FINDER (role defined in the previous stage during the design of the CL activity). Here, the student with this role will have to choose the correct word from the four options given. The word chosen must match the sign given by the SIGNER (role defined in a previous stage for this CL activity).

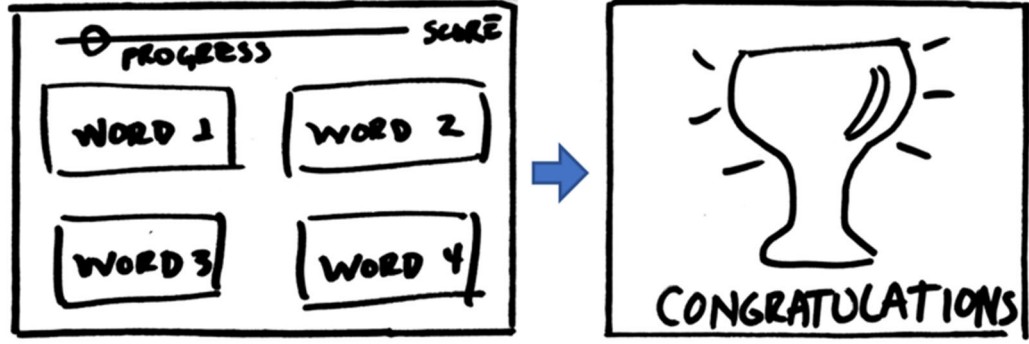

**Figure 12.** Collaborative learning screen and Achievement screen.

The content of this screen (left) is:

- A progress bar that shows how the team is doing,
- A numeric score with points obtained for every word found correctly,
- The four options to choose from (word 1, word 2, word 3, word 4).

The image on the right shows a reward once all the words are found by the team.

This activity was first carried out with sticky notes (one for the SIGNER and four for the WORD FINDER). Then, the digital prototype was implemented for Android devices and was developed to fit any screen size. In order to find out if children prefer to work individually or as a group, another activity was implemented to be done individually (level 1), and that way the collaborative learning activity is unlocked (level 2). Figures 13–20 show the final result of the prototype.

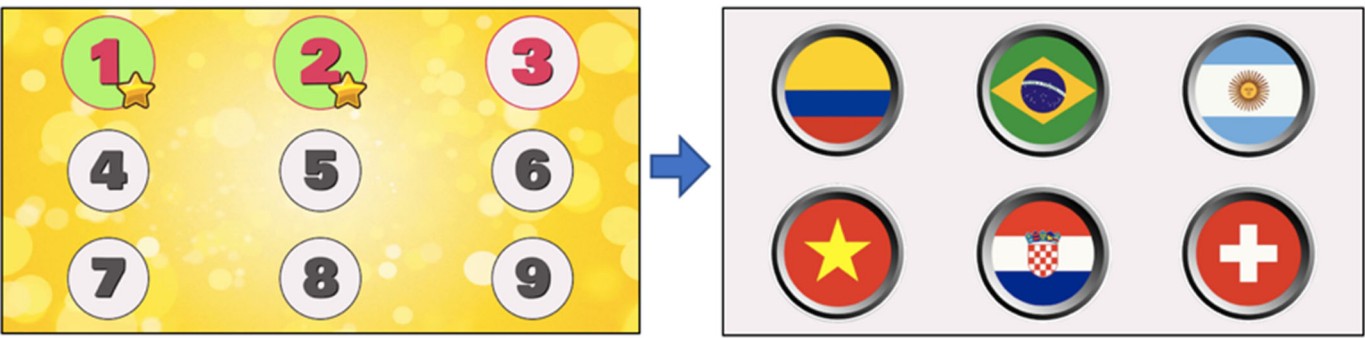

**Figure 13.** Level screen and Team Identity screen.

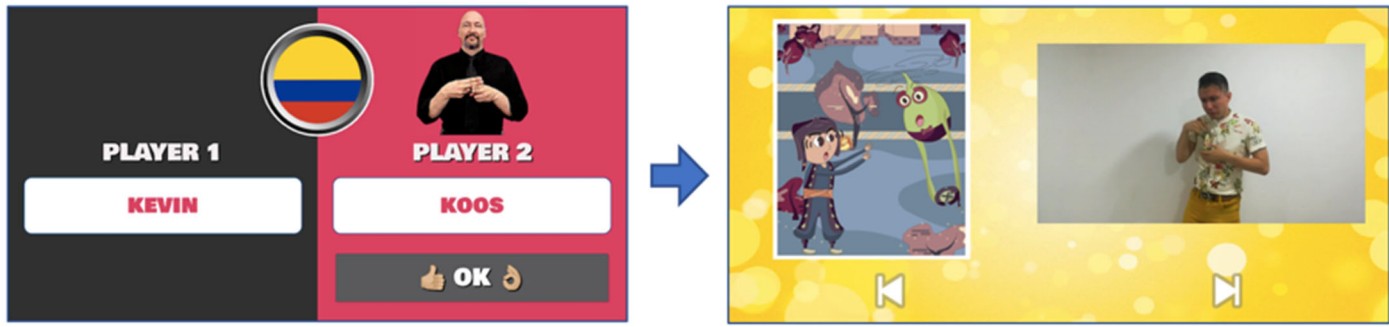

**Figure 14.** Members Names screen and Storytelling screen.

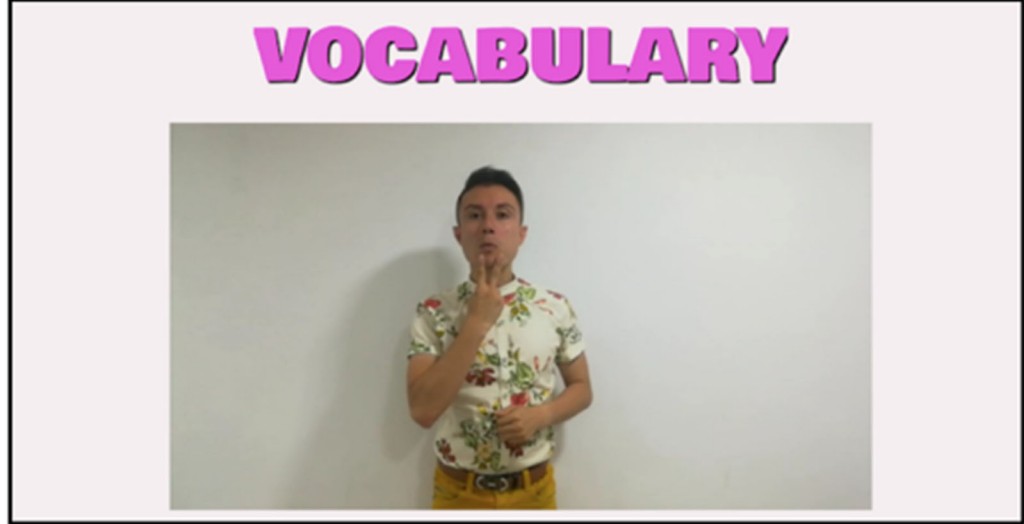

**Figure 15.** Introduction to Vocabulary.

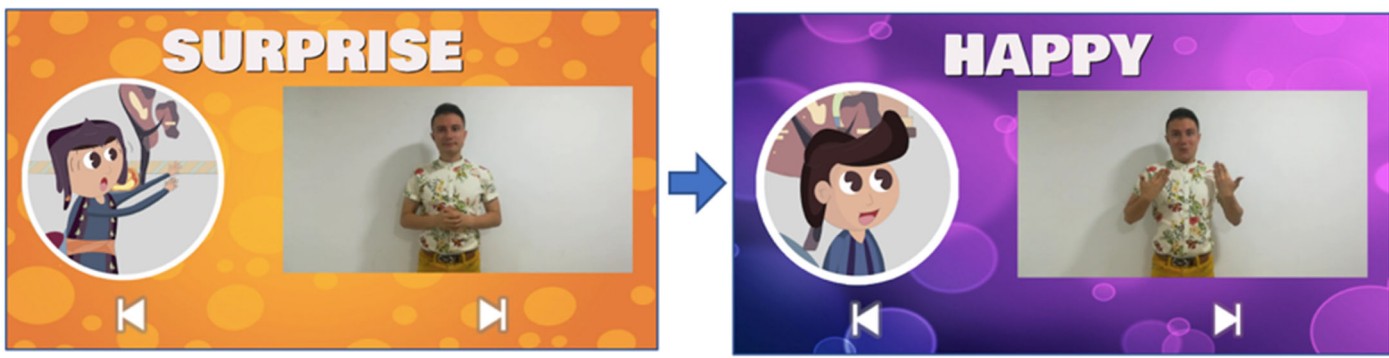

**Figure 16.** Word Training screens.

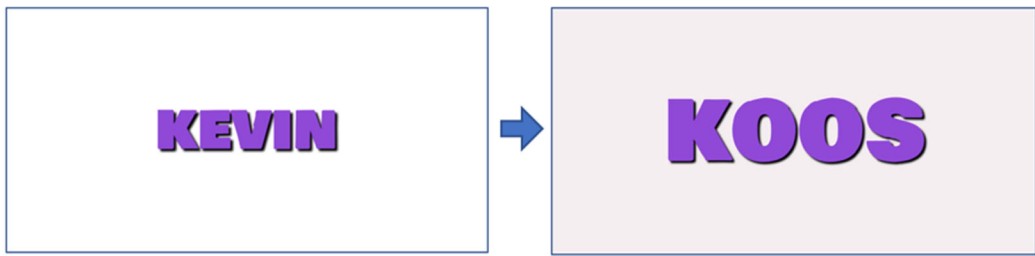

**Figure 17.** Turn screen for the CL activity.

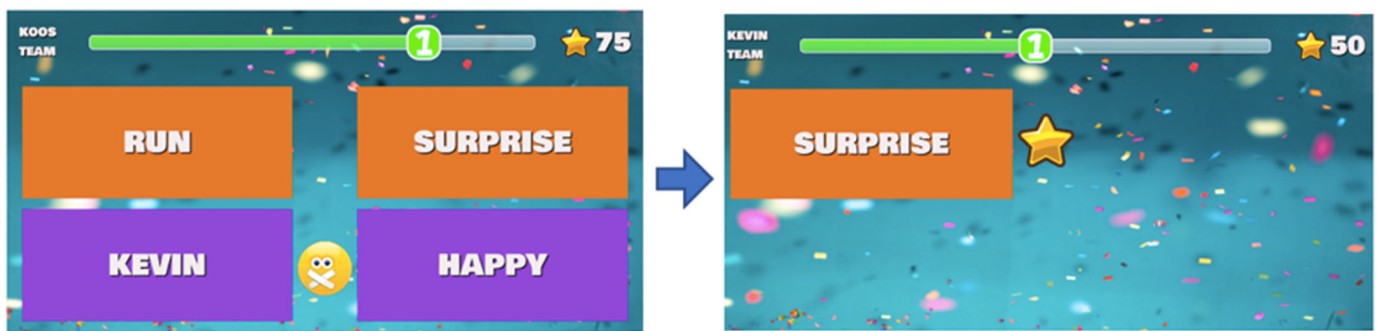

**Figure 18.** Screens showing the player selected an incorrect word (KEVIN) or the correct word (SURPRISE).

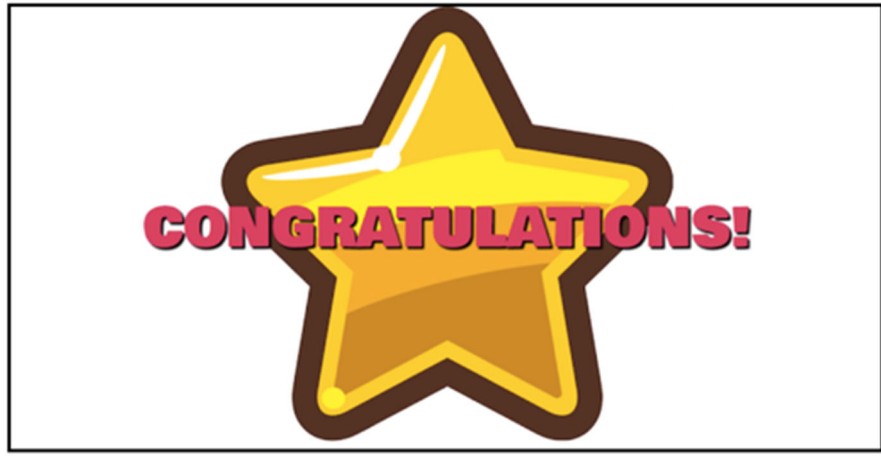

**Figure 19.** Achievement screen.

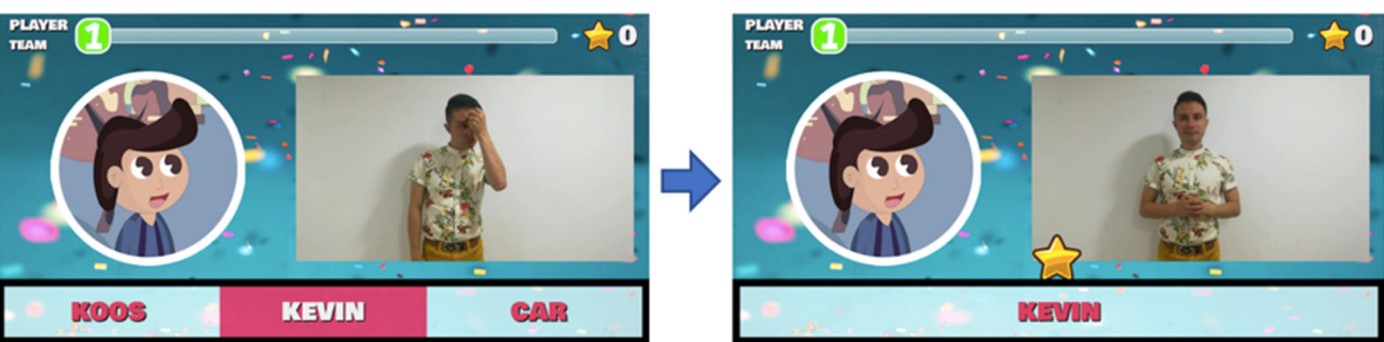

**Figure 20.** Individual Activity screen.

The image on the left in Figure 13 is the level screen where children can select the level to work in.

The content of the screen (left) is:

- A series of numbers showing the available levels (1 to 9).
- Numbers with a red and green background with a star mean that level has been completed (1 and 2).
- Numbers with a red and white background mean that level is available but not completed.
- Numbers in gray are for levels that have not been unlocked. Every time a level is completed, the next one is unlocked.

The image on the right shows the screen for team identity, where students must come to terms, and select a flag that identifies them as a team.

Figure 14 shows on the left the screen where students can type their names. This screen was not designed in the low-fidelity prototype but was designed in this prototype to give individual accountability to students, so every time they see their name on the screen, they know that part of the activity depends on them.

The content of this screen (left) is:

- Sign language on top (the sign for the word 'name').
- Text fields for every player's name.
- A button to continue.

The image on the right shows how the story is told. The content of this screen (right) is:

- An image with the scene of the story.
- A video with an interpreter telling the story in sign language.
- Two buttons to go to the previous/next scene.

Figure 15 shows the screen that indicates that a vocabulary section is about to start. The content of the screen is:

- The word VOCABULARY on top.
- A video with an interpreter signing the word VOCABULARY.

Figure 16 shows the screens where children will learn new vocabulary from the story they just read in sign language. Note how the color of the background changes depending on the kind of word. This was made to apply the Fitzgerald Key [22] strategy used in educational institutions. This strategy consists of defining a color code for every type of word (verbs in orange, nouns in brown, adjectives in purple), and this way children can learn grammar structure. The content of the screen is:

- The word to be learned on top of the screen (verb or adjective).
- An image from a scene of the story that represents the word to learn.
- A video with an interpreter signing the word to learn.
- Two buttons to go to the next/previous word.

The screen seen in Figure 17 shows the name of the student that will play the WORD FINDER role for the CL activity and thus take turns for every word they find (this was established in the design of the activity). The other student will then be the SIGNER.

The content of the screen is:

- The name of the student who will be the WORD FINDER (will hold the tablet with the four options).

The screen seen by the WORD FINDER in the CL activity is shown in Figure 18. This student will receive the word to find from his teammate (SIGNER). Then s/he will have to find the correct word. If the WORD FINDER makes a mistake, an icon will appear from behind the word s/he chose, and haptic feedback is received (no points are subtracted from the score). If the word matches the sign given by the SIGNER, the incorrect words disappear and a start appears from behind the selected word, then the score is incremented. Note how the buttons have a background color (orange or purple) to match the color-code of the Fitzgerald key. There could be words in a button with the wrong color for incorrect words. Words that are not part of the story are also included to make it easier to find the right word (taking into account this would be a basic level of the app).

The content of the screen is:

- Name of the student playing the WORD FINDER.
- A progress bar showing how the team is doing.
- A numeric score with the number of stars earned.
- Four buttons with options to select from.

Figure 19 shows the screen seen by the students when all the words were found as a team. The content of the screen is:

- A message (CONGRATULATIONS) and an image (STAR).

Finally, Figure 20 shows the activity to be performed individually by students. To access this screen, students first have to watch the story in sign language (Figure 14), then learn new vocabulary (Figure 16), and finally the individual activity.

The prototype designed meets all the requirements and information gathered in previous stages during the design of the collaborative learning activity. In the following section, the prototype is tested and evaluated by deaf children as well as by the teachers.

## 6. Case Study

The case study was carried out with deaf children from the Association of Deaf People from Valle (ASORVAL for its acronym in Spanish) and La Pamba educational institution.

### 6.1. Methodology

Twenty-four deaf children participated in a usability test, twelve from an institution in Cali-Colombia (ASORVAL) and twelve from an educational institution (La Pamba) in Popayán-Colombia. The test was carried out in both institutions. Android devices were used to run the prototype and some of the information that was collected includes demographic information, satisfaction assessment, and suggestions for improvement. No tasks were defined for this test since children prefer to explore technology by themselves (based on findings of previous case studies). Two sessions were carried out in ASORVAL (six children per session) and one session in La Pamba. Four Android tablets were used in every session, so a maximum of four children interacted with the prototype at the same time. First, the individual activity was done (level 1); once it finished, the collaborative learning activity was unlocked (level 2).

Consent forms were signed by the legally authorized representatives of the children to guarantee that all the information collected will be used only for research purposes and that no sensitive or private information would be exposed, such as names or the faces of the children. Ethical codes of the educational institutions were also followed (Appendix A).

### 6.2. Participants

Twelve deaf children from ASORVAL (Cali) participated in two different sessions (six per session). Children from the first session are in 4th and 5th grades and their ages range from 12 to 15 years old. All children are profoundly deaf and one of them has a cognitive deficit and a possible mental disorder (they will be identified as S1 to S6). Children from the second session are in 4th and 5th grades and their ages range from 11 to 15 years old. All children are profoundly deaf and five of them have other types of disorders (they will be identified as S7 to S12). S8 has a mental disorder and low vision, S9 has a cognitive deficit, S10 also has a cognitive deficit, S11 has seizure syndrome, and S12 has behavioral problems.

Twelve more children from La Pamba (Popayán) participated in one session. These children are in 2nd and 3rd grades and their ages range from 8 to 15 (they will be identified as S13 to S24). All children are deaf, S13 also has a cognitive deficiency, S15 has a mild cognitive deficiency, S17 is just learning sign language, S20 has also a cognitive and physical disability, and S24 has a cognitive deficit.

Every child answered a series of open-ended questions about the experience, the activities carried out, and the prototype in general.

### 6.3. Post-Test Questionnaire

Open-ended questions will be asked at the end of the individual activity as well as at the end of the collaborative learning activity in order to learn the children's perception of the activity and the prototype. Questions to be asked per activity are:

1.  Individual activity questions
- How do you feel after using the app? Why?
- Was the app easy to use?
- What did you like about the app?
- What did you not like about the app?
- Did you like the story?
- Would you like to play again?
2.  Collaborative activity questions
- How do you feel after this new activity with your classmate? Why?
- What did you like about this new activity?
- What did you not like about this new activity?
- Which of the activities did you like more? Why?
- Do you prefer to play by yourself or with a classmate?

### 6.4. Evaluation Techniques

The techniques that will be used are the same proposed in the DesignABILITY framework [7]: direct observation, questionnaires, and smiley-ometer. The principles given by the framework to evaluate collaboration will be also used.

The first click (tap) test was used to identify whether or not children could locate the first interactive element of every screen to move forward in order to complete the activity. While this evaluation does not give any feedback about the use of the CollabABILITY cards, it was important for us to identify usability issues in both, the individual and collaborative activities.

### 6.5. Metrics

Some metrics to evaluate collaboration were selected from the ones given in the framework. Due to the simplicity of the activity, only two metrics were defined.

- Number of errors (if they are not related to usability but to teamwork)
- Solution to the problem (important to know if they achieve the main goal as a team)

### 6.6. Usability Test with Deaf Children (ASORVAL and LA PAMBA)

The usability test was first carried out at ASORVAL with twelve children (four children at a time due to the number of available devices). The first activity was done individually

and the second one was done collaboratively. Six children participated in the first session and the remaining six in the second session (Figure 21).

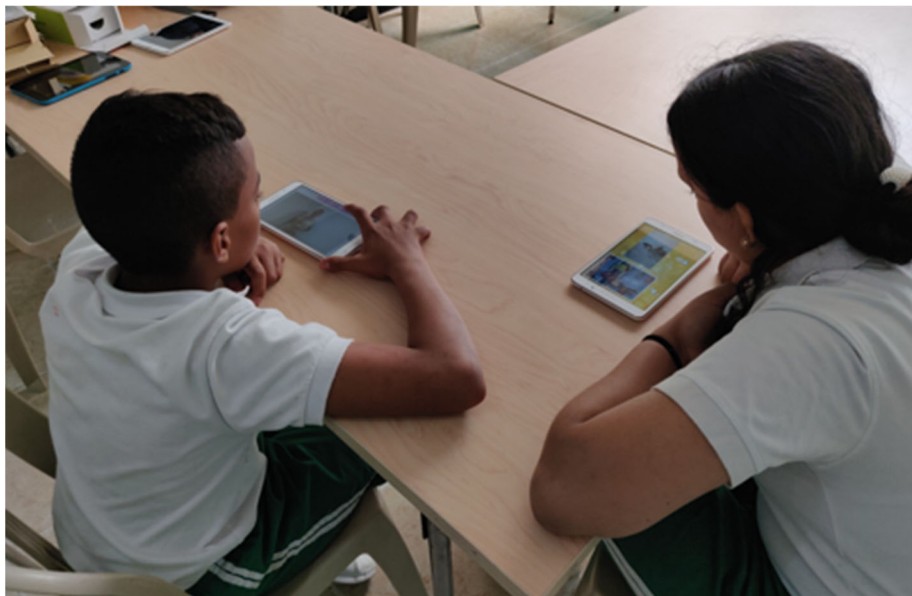

**Figure 21.** Children using the prototype at ASORVAL.

No instructions were given to the children, the tablets were handed into the *level screen* with only the first level unlocked.

For the individual activity, the *first tap test* passes if children start using the app by tapping the number 1 (first level) (Figure 22).

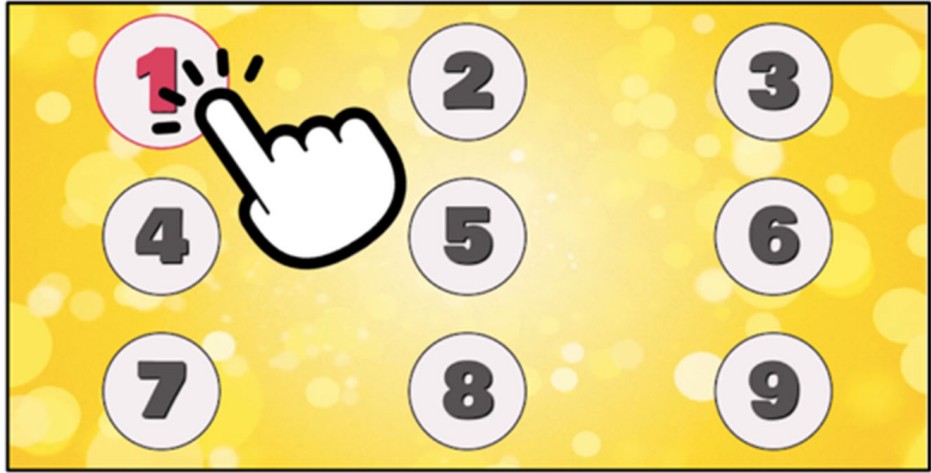

**Figure 22.** First tap test (level 1).

For the second screen (storytelling screen), the *first tap* passes if children change the scene and move forward by tapping the right button with the *next* icon (Figure 23).

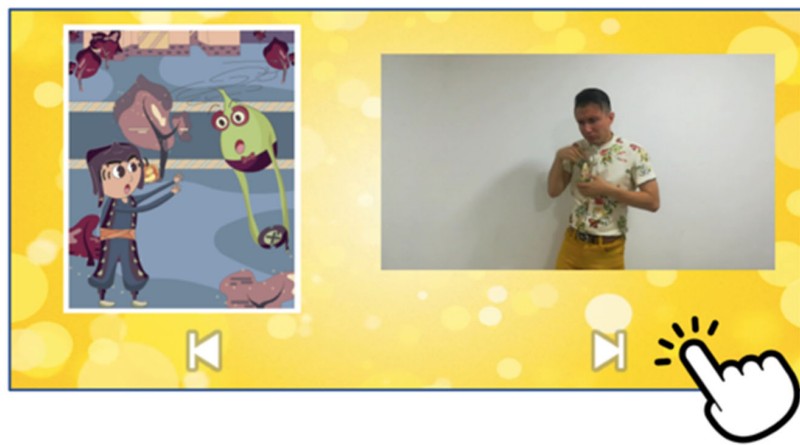

**Figure 23.** First tap test (storytelling screen).

For the third screen (vocabulary screen), the *first tap* passes if children move to the next word by tapping the right button with the *next* icon (Figure 24).

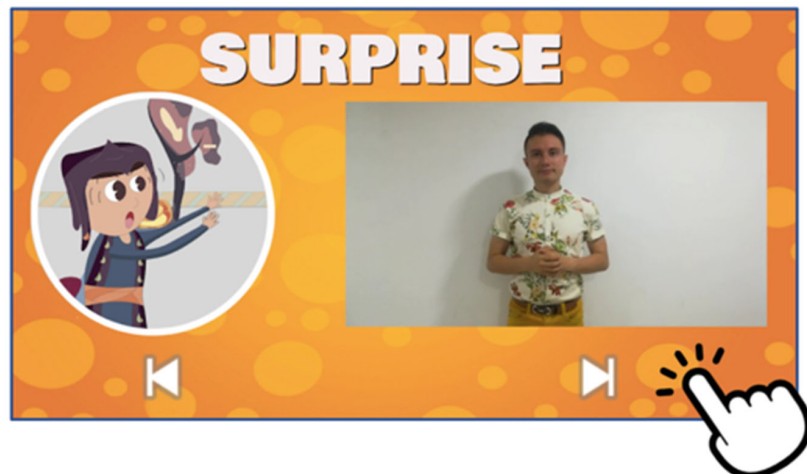

**Figure 24.** First tap test (vocabulary screen).

Finally, in the last screen (individual activity screen) the *first tap* passes if children press any of the three words to choose from (for this screen, a hint to choose the correct word is given only for the first word) ((Figure 25)).

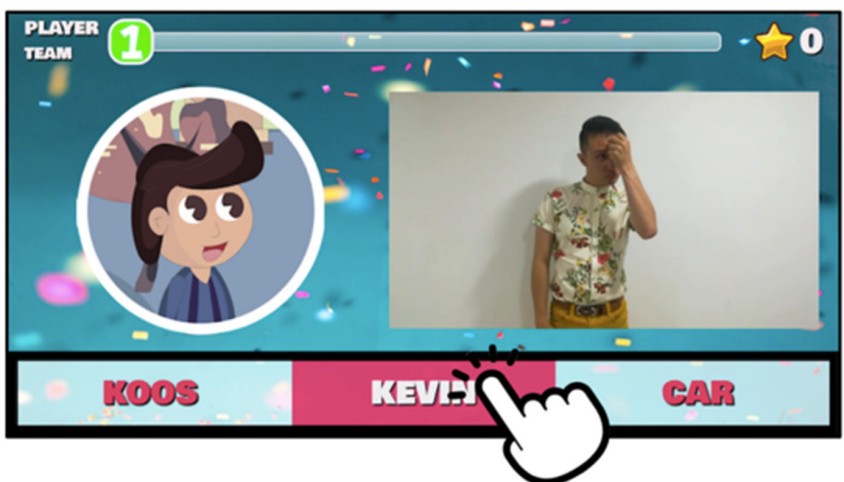

**Figure 25.** First tap test (individual activity screen).

For the collaborative activity, children were split into groups of two children. Once again in the level screen, the *first tap test* passes if children choose the number two (second level) which is now unlocked after finishing the individual activity (Figure 26).

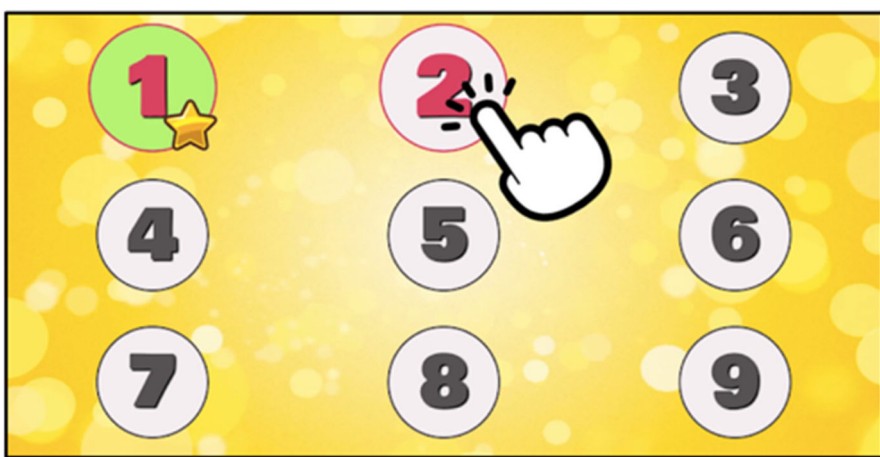

**Figure 26.** First tap test (level 2).

The second screen shows the team identity options (flags) to choose from (see Figure 13).

The third screen gives team members the option to type their names. The first tap test passes if children tap on the text field to make the keyboard appear (see Figure 14).

The fourth screen shows new vocabulary to learn from the story. The first tap test passes the same way as seen in Figure 24.

Finally, for the collaborative learning activity, the first tap test passes if THE WORD FINDER selects any of the four options to choose from (hopefully the correct one). See Figure 18.

Once children ended every activity (individual and collaborative), questions were asked to let them express how they felt after using the app.

The usability test was then carried out in La Pamba educational institution with twelve children (four children at a time) in 1 session (Figure 27).

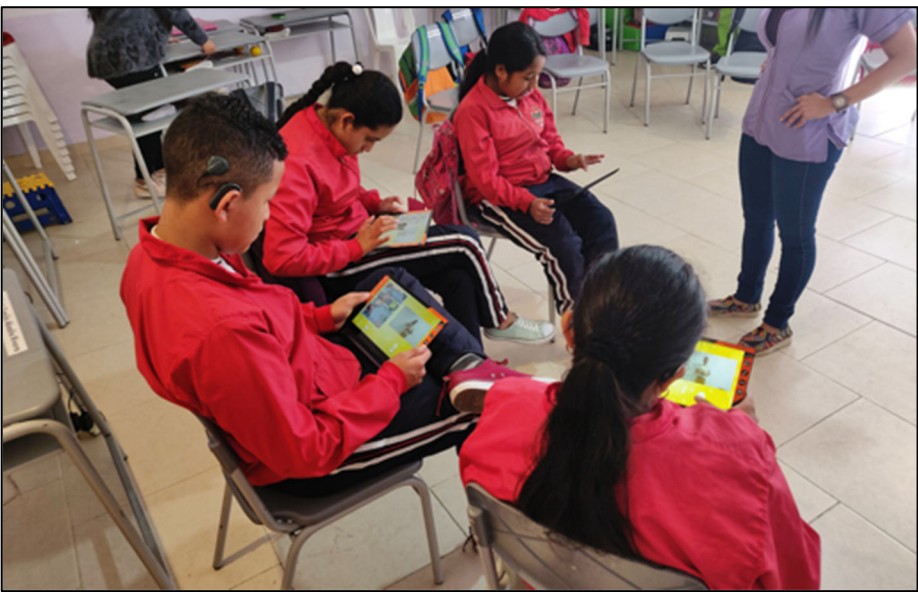

**Figure 27.** Children using the prototype at LA PAMBA.

6.6.1. Results of the Usability Test

For the *first tap* test on each screen, the results were satisfactory. For the level screen, all children identified the available levels in both activities. They also identified that level number 2 was available after finishing level 1. For the storytelling screen, one child from session 1 in ASORVAL and two children from session 2 in ASORVAL had trouble identifying the button to move forward on the story and asked for help. None of them had trouble identifying the first interactive element of the rest of the screens (team identity, individual activity screen, and collaborative activity screen).

Two metrics were selected to evaluate this CL activity, one of them is the number of errors made by each team. Twelve teams of two children were created (G1 to G12) and the number of errors made by each of them is shown in Figure 28.

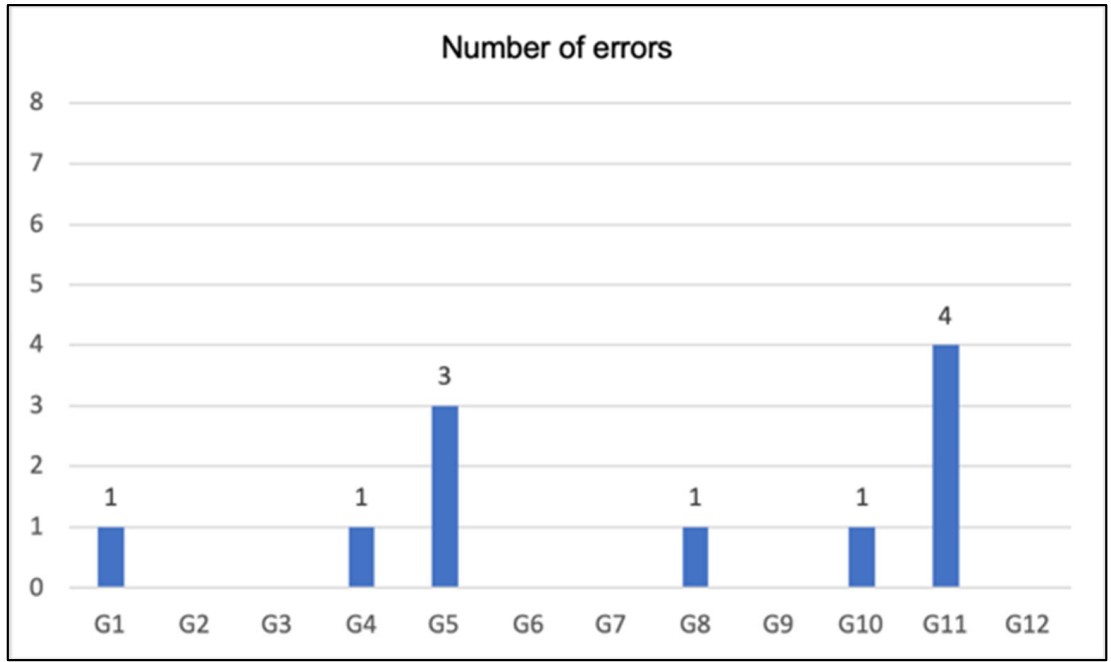

**Figure 28.** Number of errors made by each team.

Two groups made a significant number of errors (G5 from ASORVAL and G11 from LA PAMBA) compared to the rest of the groups. These errors were not related to usability issues in the design, but mistakes made during the activity due to communication between team members.

The other metric was the solution to the problem, which was achieved by all groups, some with difficulties due to understanding the activity of one of the team members, but in the end, the main goal was achieved by all groups.

A short survey was taken for each child after each activity. Due to the extent of all the answers given by the twenty-four children, the information with these results is shown in Appendices A and B. A summary of these answers is given in this section and discussed in the next one.

The six questions asked of the children after the individual activity were:

- Q1: How do you feel after using the app and why?

All the children felt either good or happy. Some of the reasons were because they understood what they had to do; they also had fun and some of them enjoyed reading the story in sign language.

- Q2: Was the app easy to use?

For Q2, three children did not answer, one of them said it was difficult because it was hard to learn the new words for her. Twenty children said the app was easy to use; some of them expressed that sign language was helpful and they easily understood what they had to do.

- Q3: What did you like about the app?

One child did not answer, and the rest said they liked either the story, the instructions in sign language, the vocabulary, the haptic feedback, or the way they learned with it.

- Q4: What did you not like about the app?

Three children did not answer, and the remaining twenty-one said they liked it, so there was nothing to say about this question.

- Q5: Did you like the story?

One child did not answer, and the rest said they enjoyed the story; some of them said this was because they understood the sign language.

- Q6: Would you like to play again?

All the children wanted to continue using the app.
The five questions asked to the children after the collaborative activity were:

- Q7: How do you feel after this new activity with your classmate and why?

All children felt either good or happy, for some it was because they played with a partner, for others it was the sharing, and for others, it was the new vocabulary.

- Q8: What did you like about this new activity?

One child did not answer, while the remaining twenty-three gave different opinions on what they liked, for instance, playing with a classmate, helping each other, the new signs and vocabulary, the rewards, the feedback (when they were right or wrong), and some enjoyed feeling in some kind of competition.

- Q9: What did you not like about this new activity?

Three children did not answer this question, the rest said they liked it.

- Q10: Which of the two activities did you like more and why?

Two children liked the first activity more without giving reasons, while the other twenty-two children preferred the second activity. The reasons included that it was more fun, and they prefer working with friends.

- Q11: Do you prefer to play by yourself or with a classmate?

Two children prefer to do the activities by themselves (the same two children who liked the first activity more) and one of them said it was because her partner did not understand the activity. The rest of the children prefer to do the activities with a peer.

### 6.6.2. Direct Observation and Collaborative Learning Evaluation

Through direct observation, it was possible to identify not just that the activities designed were enjoyable and usable, but also how the collaboration worked for deaf children. During the collaborative learning activity, most of the students understood their roles during the activity and how to use the resources given to achieve a common goal. Only two students (from different groups G5 and G11) had difficulties understanding what they had to do (due to cognitive issues associated with deafness) which led their groups to be those with a greater number of errors, while the frustration of their teammates was evident. While frustration is something we want to avoid in a learning activity, something positive that came out of these situations was that despite the frustration, the students who did understand the activity did everything they could to explain it to their teammates (with cognitive issues), which in the end is part of the collaborative learning process we aim to achieve.

Students stuck to their roles and used the information and resources given to each other in order to achieve the common goal. The rewards and feedback given during the activity

were helpful in increasing motivation during the activity. These positive interdependences, when applied properly, assured collaboration among team members.

6.6.3. Discussion

All twenty-four children enjoyed using technology as part of the learning process. Some of them had little trouble identifying the button to move forward on the storytelling screen, but once it was learned they easily identified it in the vocabulary screen.

In general, they enjoyed everything about the app, they said it was easy to use and helpful. Some of the results to highlight are that children enjoyed feeling the haptic feedback (vibration) when they selected the wrong option in both activities. Additionally, they enjoyed having sign language as support for the activities.

Some of the findings from direct observation are that for some teams, it was not easy to come to terms when deciding which flag would identify them; in most cases, one of the kids just pressed the one he liked before asking his peer. Additionally, some children spelled every new word they learned.

Most of the answers were similar among children, except for the last two questions, where two children said that they preferred to work by themselves rather than with a partner. It is no coincidence that both answers come from the two children who did the collaborative activity with a classmate who had trouble understanding the activity (due to his/her cognitive issues), and that it made them feel like they did not have support from their teammates to achieve the main goal; this could also be seen in Figure 28, which shows that those groups made more mistakes than the rest of the groups and thus it took longer to complete the activity. However, these children were especially willing to help their classmates whenever they had trouble understanding the activity.

It is then important to take into account how children with different academic levels (and with additional difficulties like cognitive issues) are going to collaborate and how those with higher academic levels help their peers to achieve their individual goals and thus achieve the main goal as a team without the feeling that they are in some kind of disadvantage. Activities with these kind of challenges must be carefully designed in order to increase the motivation of all team members and promote learning for all of them even at different levels. Fortunately, the activity was designed by applying some positive interdependence principles stated [21] by Laal and affordances proposed by Jeong, and Hmelo-Silver in [5], such as rewards, shared resources and communication, which lead to a nice experience for students regardless of the aforementioned issues.

**7. Evaluation of the Prototype (by Educators)**

After the usability test with children, educators who participated during the design of the activities and the prototype, took a short online survey to evaluate the prototype. The questions asked are:

- Q1: What do you think about the prototype developed?
- Q2: How do you think this prototype could be improved?
- Q3: Do you think this kind of tools could support your teaching process? Why?
- Q4: The prototype is useful to develop literacy skills in deaf children (1 to 5)
- Q5: The prototype is useful to promote collaborative learning (1 to 5)

*7.1. Results*

Responses to the questions are:
What do you think about the prototype developed?

- I think this prototype is a great way to involve technology in the classroom that is not distractive but supportive for deaf children.
- The prototype is a great idea to support learning a second language for deaf children, but there are not enough activities to see significant results.
- I love the idea behind the whole design process, involving storytelling is great to engage children and the use of sign language all over the app is very helpful for them.

- This prototype meets the needs of the children and mine too as a teacher. The way it guides children from the story to the learning activities is very clever.

  How do you think this prototype could be improved?

- Increase the number of levels and make the activities last longer to keep children motivated.
- As a prototype, the number of implemented levels is not enough, it needs more stories and activities.
- The interpreter (sign language) in the videos, should use a shirt with a solid color instead of a shirt with images or figures that could distract children when doing signs.
- The implementation of more activities, both, individual and collaborative. Do you think this kind of tools could support your teaching process? Why?
- Definitely, there are no applications that cover this literacy process in such a detailed way. At least not in Spanish. I hope the 'prototype' becomes a complete application full of stories and activities.
- Sure. It is a great way to complement the activities carried out during the school year.
- Yes, but I would also need access to this technology like tablets, unfortunately, there are not enough resources to have these kinds of devices.
- Yes, actually, it could be very helpful for children to have this application at home and improve their literacy skills on their own.

  The prototype is useful to develop literacy skills in deaf children (1 to 5) (Figure 29).

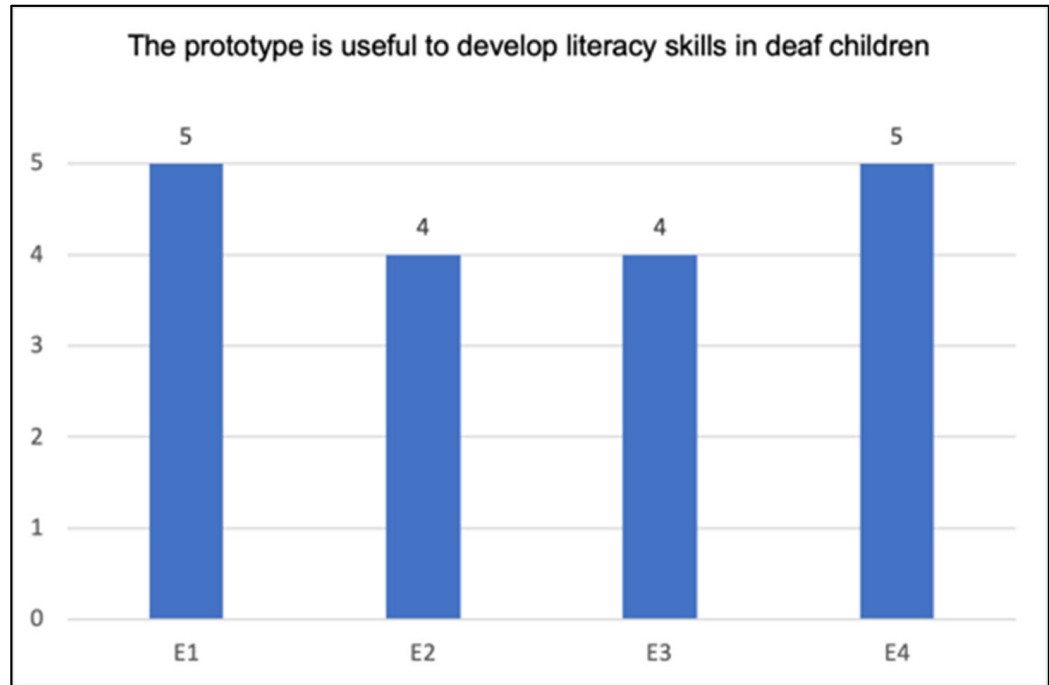

**Figure 29.** Usefulness of the prototype to develop literacy skills.

All educators (E1 to E4) rated the prototype (5-Likert scale) in terms of usefulness to develop literacy skills. The average rate is 4.5 out of 5.

The prototype is useful to promote collaborative learning (1 to 5) (Figure 30).

All educators (E1 to E4) rated the prototype (5-Likert scale) in terms of usefulness to promote collaborative learning. The average rate is 4.25 out of 5.

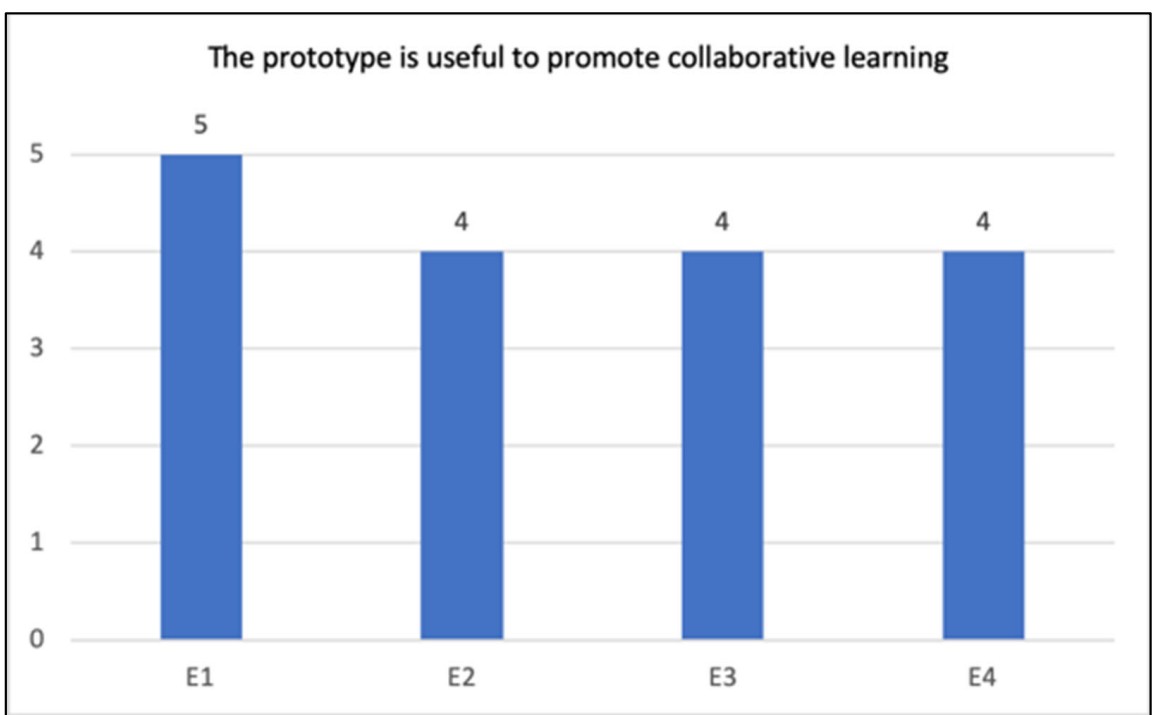

**Figure 30.** Usefulness of the prototype to promote collaborative learning.

*7.2. Discussion*

The questions asked aimed to find out how educators perceive this kind of technology and how useful they think it could be to promote collaborative learning among their pupils and support their teaching process.

All educators have a great impression of the work done with the prototype. They all think it is a good way to make use of technology in the classroom and see it as a resource to improve their teaching process. Unfortunately, there is a lot more to be done in order to have a complete application to support all learning goals (for a given grade), but according to the results of all the evaluations done, we are on the right track to arrive there.

They do think that the complete version of the prototype could help deaf children to develop literacy skills and promote collaborative learning among them. Something interesting about the collaborative learning activity that teachers were not expecting was how children with a high academic level were willing to help their classmates with cognitive issues. Even though they felt a little bit frustrated because their team was not doing very well, it did not restrain them from helping their peers just as Smith and Macgregor stated in [1]: "In collaborative endeavors, students inevitably encounter difference, and must grapple with recognizing and working with it". This also agrees with García et al. in [4] where they state that CL increases student learning, motivation and the repercussions of collaborative learning on students with learning disabilities.

## 8. Conclusions and Future Work

Designing tools to support teaching to children with disabilities is quite challenging for designers or software developers as they usually do not know about the special needs these children have and there is no specific-purpose methodology to be followed for particular learning goals, needs and disabilities. Through the DesignABILITY framework and more specifically through the CollabABILITY cards/templates, it is possible not just to design tools aimed at deaf children but also to promote collaborative learning supported by computers.

In this study, the CollabABILITY cards/templates were used to design activities aimed at deaf children and a particular learning goal such as literacy, but it may be also used or adapted to design CSCL activities for children with other types of (dis) abilities. It was very

important to have a multidisciplinary team to design and evaluate the cards/templates as these are meant to be used by designers/developers and educators.

It is important to mention that it is necessary to structure activities to convey a real collaboration, due to just making a group of people work around a task does not guarantee real collaboration and participation. As part of this research, the design of activities where aspects like positive interdependence, equal participation, and individual accountability (fundamental aspects of collaborative learning processes) can be integrated along with the storytelling process through the CollabABILITY cards/templates. Positive interdependences were successfully implemented in the designed tool and according to children and educators' perception, they do promote work among peers even with different academic levels, actually, children enjoyed the CL activity more, especially those with associated cognitive issues who found in their classmates the necessary help that was not provided in the non-collaborative activity. These cards/templates were validated by both, designers and educators, and the results during the design of collaborative learning activities demonstrate how useful these cards are, even for teachers who want to implement a CL strategy with the use of available technology in the classroom.

Collaborative learning strategies have been proven to promote different skills in learners, but its implementation is not easy, even with technology as a mediator. Tools should be designed taking into account the differences between learners, especially those with some type of disability. By creating systems with the DesignABILITY framework that support CSCL, children with disabilities (deafness in this study) may have the opportunity to collaborate not just with other deaf peers, but also with normal-hearing children.

The digital version of the cards/templates will be extended to other platforms like iOS devices and desktop computers.

The developed prototype will be completed by adding more levels (learning activities) to support teaching of more learning goals. This completed version will be tested again with deaf children from Popayán and Cali in Colombia. The support of other languages (English, Portuguese, American Sign Language, British Sign Language, Brazilian Sign Language and Portuguese Sign Language) is necessary in order to provide a multi-language tool that supports literacy teaching to deaf children from other countries and cultures.

**Author Contributions:** Conceptualization, L.F.-A. and C.A.C.; methodology, L.F.-A., S.C., A.S. and C.A.C.; software, L.F.-A.; validation, L.F.-A.; formal analysis, L.F.-A., S.C., A.S. and C.A.C.; investigation, L.F.-A., S.C., A.S. and C.A.C.; resources, L.F.-A.; data curation, L.F.-A.; writing—original draft preparation, L.F.-A.; writing—review and editing, L.F.-A., S.C., A.S. and C.A.C.; visualization, L.F.-A.; supervision, S.C., A.S. and C.A.C.; project administration, L.F.-A. and C.A.C.; funding acquisition, L.F.-A. All authors have read and agreed to the published version of the manuscript.

**Funding:** This research was funded by Institución Universitaria Antonio José Camacho. Code: PI-0922. Resolution No. 072.

**Institutional Review Board Statement:** The study was conducted in accordance with the Declaration of Helsinki and approved by the Institutional Review Board (or Ethics Committee) of Institución Universitaria Antonio José Camacho (approved on 1 April 2019) for studies involving humans.

**Informed Consent Statement:** Informed consent was obtained from all subjects involved in the study or their parents/relatives when necessary.

**Data Availability Statement:** Not applicable.

**Conflicts of Interest:** The authors declare no conflict of interest. The funders had no role in the design of the study; in the collection, analyses, or interpretation of data; in the writing of the manuscript; or in the decision to publish the results.

## Appendix A. Ethical Considerations

*Appendix A.1. Association of Deaf People from Valle (ASORVAL)–Ethical Code*

1. Guarantee children and adolescents the necessary care and attention for their integral development, both physical and cognitive, relational, emotional, spiritual, and ethical in accordance with the established care process of each modality.

2. Prevent the occurrence of situations of abuse, discrimination, mistreatment, stigmatization or any action or omission against the fundamental rights of children and adolescents.

3. Ensure the timely identification of situations that endanger the life and physical, emotional, and mental integrity of children and adolescents you are in charge of, for as long as they are under your care or responsibility. In case of knowledge about possible abuse, you should inform the competent authority immediately.

4. Have respect and reserve for the life history of the children and adolescents in charge, without exploring about it or trying to deepen specific information, that is outside contributing to the restoration of rights and that does not obey the best interests. The information recorded in these stories is restricted and must be kept under absolute reserve and confidentiality.

5. Respect the privacy and the right to privacy of children and adolescents under your care.

6. Establish a communication through healthy, assertive, kind, and respectful messages to the children and adolescents in charge. In the case of indigenous children/adolescents, the ethnic approach in the section on differential approach of this document must be taken into account.

7. Engage in the direct care of children and adolescents, without delegating their attention, or leaving them in charge of people who are not part of the mode of care, or the family, unless duly authorized by the administrative authority in charge.

8. Share with children and adolescents, activities within the framework of respect, trust, empathy, and good treatment. Establish relationships characterized by equity, justice, and solidarity and non-discrimination.

9. Assume a role of consideration and respect towards children and adolescents as subjects of rights and demand it equally from those who interact with them.

10. Refrain from behaviors or expressions of discrimination, rejection, indifference, stigmatization, or other treatment that affects the mental, emotional, or physical health of children and adolescent.

The following are the actions that expose children and adolescents to non-observance, threat, or violation of rights and are considered infringements of the ethical code:

(a) Impose sanctions or punishments that attempt against physical or mental integrity and the development of the personality of children and adolescents.

(b) Discriminate by race, sex, gender, religion, sexual orientation, physical, mental disability, or by any other condition.

(c) Physical, verbal, or psychological abuse or neglect in the care of children or adolescents.

(d) Deprive children totally or partially of food or create delays in the meal schedules of children and adolescents under your responsibility or care.

(e) Use in the preparation of food, ingredients that, prior to technical studies, the ICBF or administrative authority considers harmful to the health of children or adolescents.

(f) Deprive of the supply of medications in accordance with those formulated, use medications whose date of expiration has been met or supply medications that have not been formulated by a doctor legally authorized for the exercise of the profession, to children and adolescents who are under your responsibility or care.

(g) Not to carry out the necessary and pertinent steps in the timely provision of the health service when required by a child or adolescent under your responsibility or care.

(h) Deny the provision of personal endowment (bed, mattress, bedding, clothing, toiletries, educational or recreational material, or provision according to the cultural practices of ethnic groups) to children or adolescents under your responsibility or supply inadequate equipment or in poor condition for its use.

(i)     Exclude children or adolescents from academic training or recreation programs, based on race, gender, sexual orientation, disability, or any other discriminatory situation.

(j)     Deprive children or adolescents the right to have visitors or to communicate, with family or relatives, except in cases in which the competent administrative authority has justified it.

(k)     Permit and tolerate acts of abuse or harassment among children and adolescents, who interact in the different programs.

(l)     Deliberately omitting the complaint or communication of acts of abuse, harassment or sexual abuse of children or adolescents before the competent authority or authorities. Likewise, not taking any action to protect children or adolescents against such abuses.

(m)   Use children or adolescents for the purpose of economic exploitation or in jobs that threaten their physical and emotional health or personal integrity.

(n)     Failure to comply with safety and disaster prevention norms or any risk to the health and integrity of children or adolescents.

(o)     Failure to comply with safety norms in the transport of children or adolescents, in accordance with the provisions of the traffic code and other rules related to school transportation.

(p)     Give way out of the care process or suspend the attention of children or adolescents, without the authorization of the defense interdisciplinary technical team of family or of the competent authority in charge of the case.

(q)     Hide, delay, or partially deliver to the ICBF the information about children or adolescents, which would eventually lead to a change of measure or decision making in the framework of the process of attention.

(r)     Not having the documents established by the ICBF and the health sector or performing inappropriate practices.

*Appendix A.2. Consent Form*

I agree to allow my child to participate in the usability study conducted by researcher Leandro Flórez Aristizábal, Ph.D. student at the University of Cauca.

I understand and allow the video recording of the activity by Leandro Flórez, bearing in mind that my child's face will not appear in such material. My child's identity will remain completely anonymous and that the collected data will be used for research purposes only.

I understand that the material obtained (photos and video) will be destroyed once the data is analyzed by the researcher.

Participation in this study is voluntary and I agree to let Leandro Flórez or teacher in charge know about any doubts or concerns that my child or I might have about the activity to be carried out.

Please sign below stating that you have read and understood the information on this form and that any questions you have had about this study have been answered

Date:_________

Name of child: _________________________________________________

Child's parent or representative name: _________________________________

Signature: _____________________________________________________

Thanks!

*Appendix A.3. Consent Form (Signed)*

Investigador: Leandro Flórez Aristizábal
Est. Doctorado de la Universidad del Cauca
Docente Investigador de la Institución Universitaria Antonio José Camacho

## Formato de consentimiento

Acepto que mi hijo participe en el estudio de usabilidad realizado por el investigador Leandro Flórez Aristizábal, estudiante de doctorado de la Universidad del Cauca.

Entiendo y permito la grabación de video de la actividad por parte de Leandro Flórez, teniendo en cuenta que el rostro de mi hijo no aparecerá en dicho material. La identidad de mi hijo permanecerá completamente anónima y que los datos recolectados serán usados únicamente con propósitos investigativos.

Entiendo que el material obtenido (fotos y video) será destruido una vez los datos sean analizados por parte del investigador.

La participación en este estudio es voluntaria y acepto dar a conocer a Leandro Flórez o docente encargado, cualquier duda o preocupación que mi hijo o yo podamos tener sobre la actividad a realizar.

Por favor firme a continuación indicando que ha leído y entiende la información en este formulario y que cualquier duda que haya tenido acerca de este estudio ha sido respondida

Fecha: _Agosto 14 – 2019_

Nombre del niño: _____________________

Nombre del padre o madre: _____________________

Firma: _____________________

*Gracias!*

Agradecemos su participación.

**Figure A1.** Signed consent form.

## Appendix B. CollabABILITY Cards/Templates

These cards/templates were designed to help users (teachers, designers, developers) design collaborative learning activities. The templates must be used in conjunction with the cards in order to understand in detail how a collaborative learning activity is structured.

*Appendix B.1. Cards*

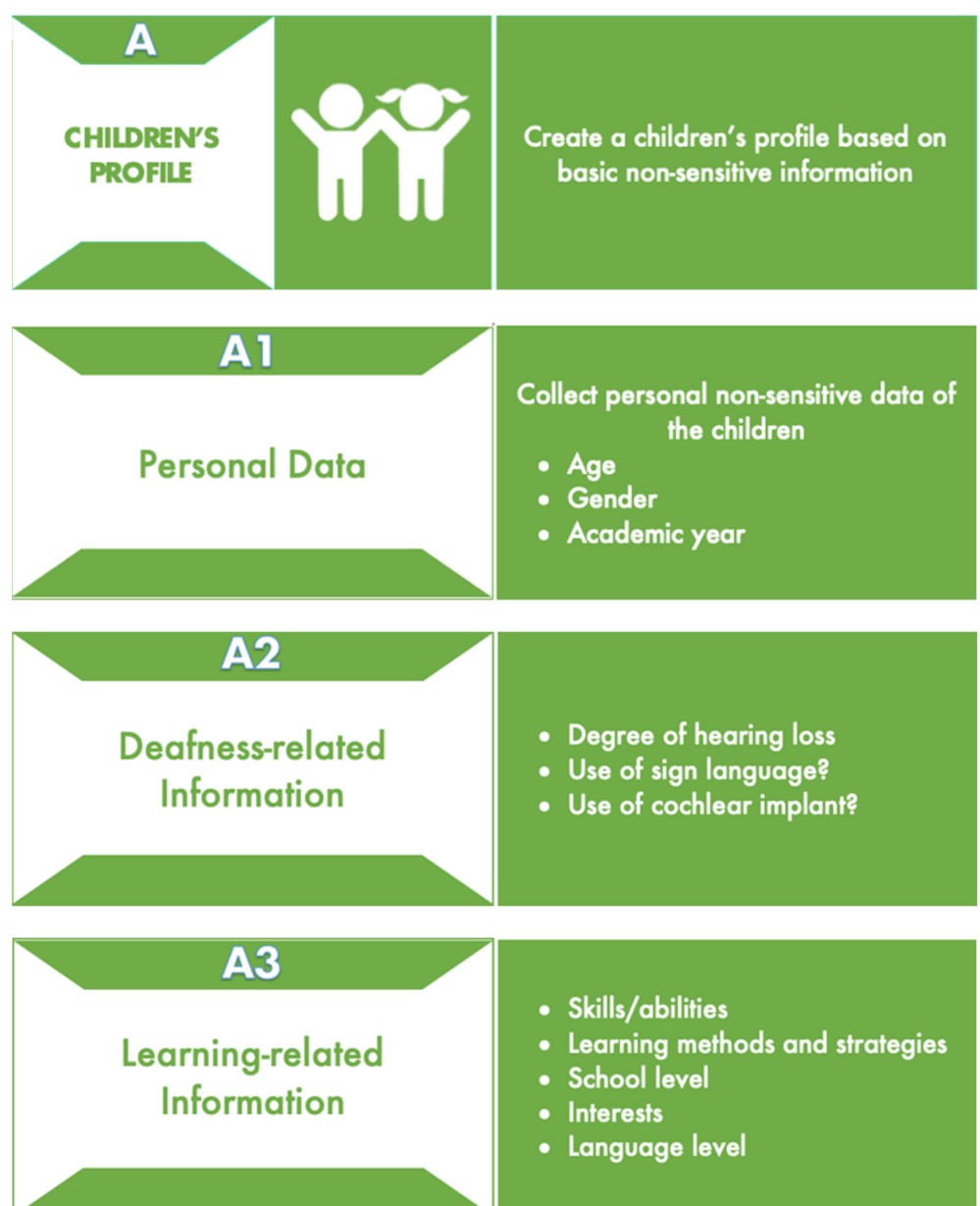

**Figure A2.** CHILDREN'S PROFILE cards.

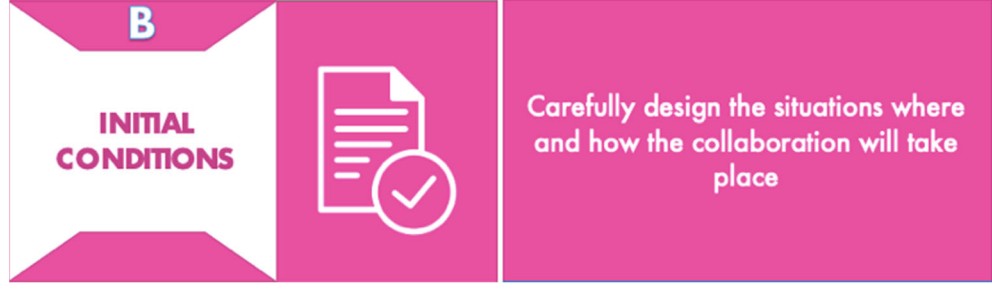

**Figure A3.** *Cont.*

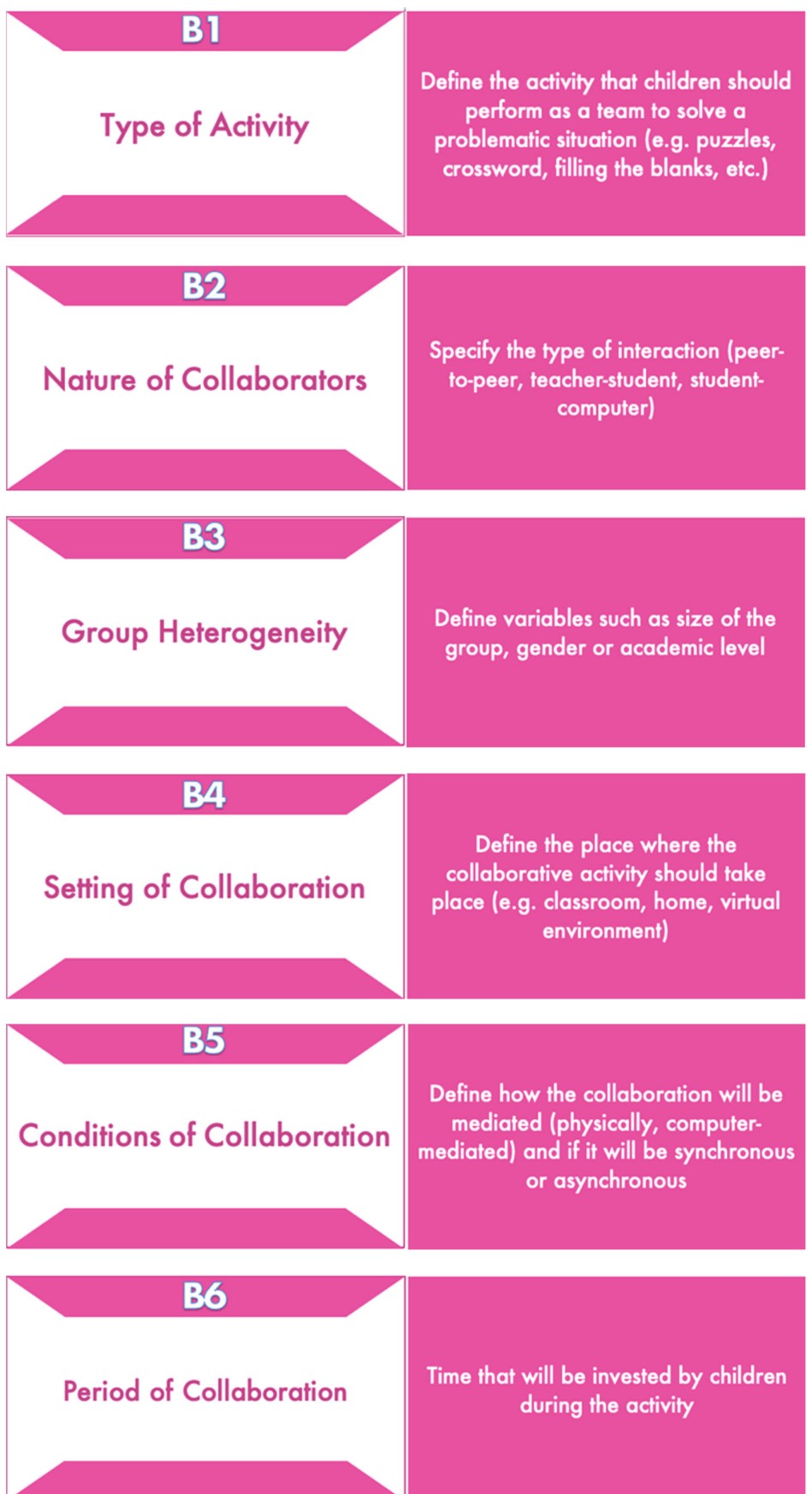

**Figure A3.** INITIAL CONDITIONS cards.

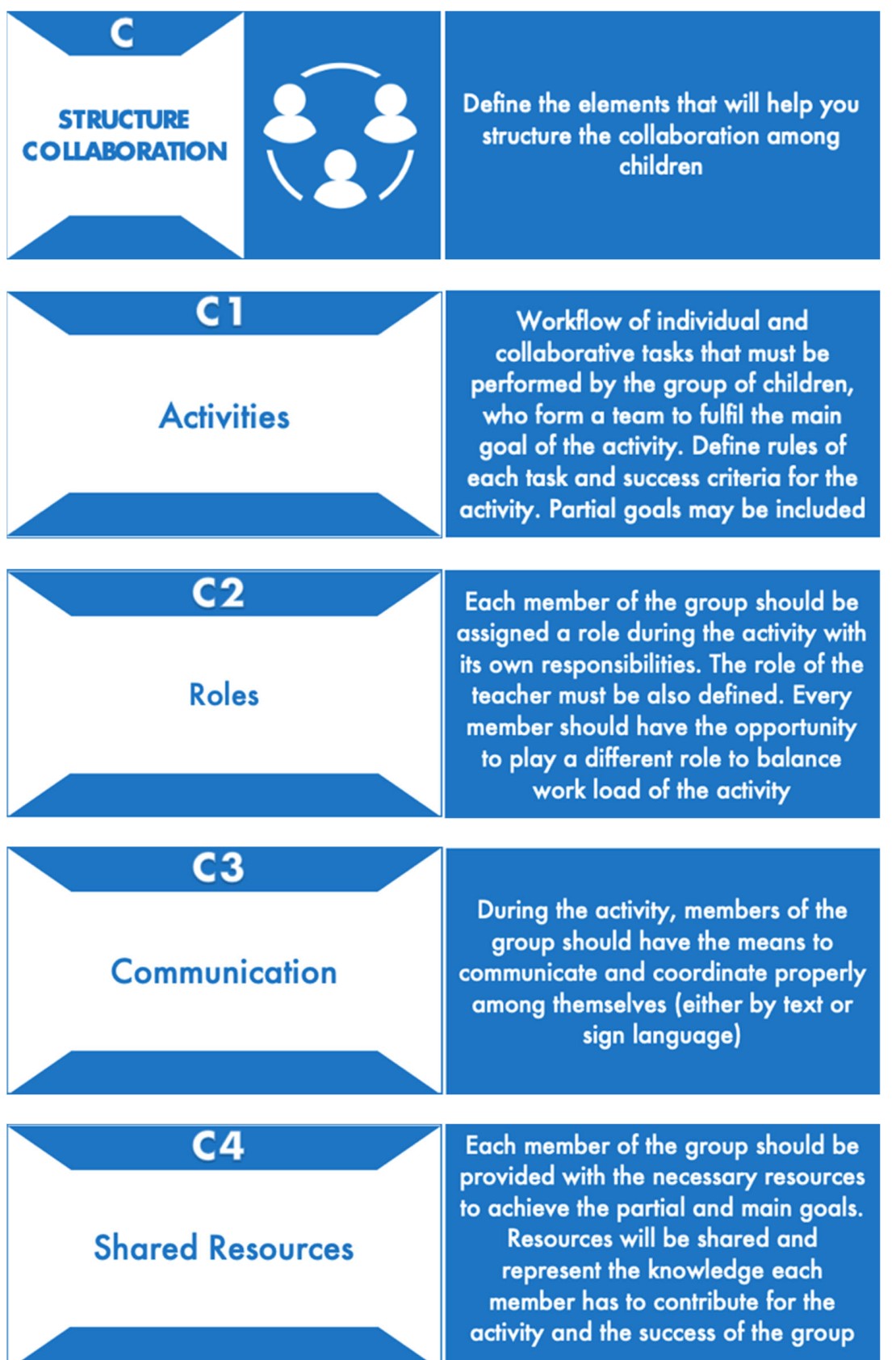

**Figure A4.** STRUCTURE COLLABORATION cards.

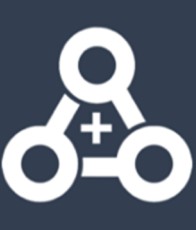

**D**

**POSITIVE INTER-DEPENDENCES**

One or more of these should be selected to guarantee collaboration

**D1**

Positive Interdependences - PI
**Game Mechanics - GM**
**Learning Mechanics - LM**

Specify the types of PI that will assure true collaboration among students and encourage them to think as "we" instead of "me". GM (if necessary) and LM should also be specified in order to promote engagement and motivation in the learning activities

**D2**

Role

Combined roles and responsibilities are required for the group to fulfill a common task

**GM – Role Playing:** The player acts out the role of a fictional character

**LM – Guidance:** Provide guidance for learning

**D3**

Identity

Makes unity and cohesion, increasing friendship and affinity through a shared identity expressed upon a common logo, motto, name, flag or song

**D4**

Goal

It is the belief that each team member can reach his or her goals only when the goals of the group are met

**GM – Progression:** The success is granularly displayed and measured through the process of completing itemized tasks.

Example: a progress bar

**LM – Assist:** Help, promote or support an equal or companion

**D4**

Goal

It is the belief that each team member can reach his or her goals only when the goals of the group are met

**GM – Goal:** Sort of victory condition. Can be broad enough to encompass any method of winning, but here refers to game-specific goals.

Example: Checkmate of a king in chess

**LM – Self - regulate:** Focus attention on one's own progress and cannel this towards achieving a goal

**Figure A5.** *Cont.*

## D4

### Goal

It is the belief that each team member can reach his or her goals only when the goals of the group are met

**GM – Cooperative play:** Encourages players to work together to beat the game. There is little or no competition between players. Either the players win the game, or all players lose it

**LM – Collaborative:** More than one learner participates in a common learning activity to pursue a common goal

## D5

### Environmental

A physical environment that unifies the members of a group in which they work

**LM – Situate:** Position learning in the context in which it is to be applied

**LM – Discover:** Gain understanding and solve problems by exploring/interacting with and manipulating the environment

## D6

### Resource

Each individual has only a part of the information, resources or materials needed for his/her task. Therefore, the resources should be combined in order to accomplish the shared goal

**GM – Communal discovery:** An entire community is rallied to work together to solve a problem/challenge. Immensely viral, a lot of fun

**GM – Cascading information:** Information should be released in the minimum possible snippets to gain the appropriate level of understanding at each point during a game narrative

## D6

### Resource

Each individual has only a part of the information, resources or materials needed for his/her task. Therefore, the resources should be combined in order to accomplish the shared goal

**GM – Resource management:** The games' rules determine how players can increase, spend, or exchange their resources (tokens, money, etc.). The skillful management of resources under such rules allows players to influence the outcome of the game

**LM – Connect:** Build knowledge by connecting information

## D7

### Task

The organizing of the group works in a sequential pattern. When the actions of one group member have been accomplished, the next team member can proceed with his/her responsibilities

**GM – Turn:** Segment of the game set aside for certain actions to happen before moving on to the next turn, where the sequence of events can largely repeat

**LM – Master:** Proceed step by step, completing learning of one aspect before tackling a more difficult/complex one

## D8

### Outside Enemy

Putting groups in competition with each other. Group members feel interdependent as they do their best to win the competition

**GM – Micro leader - boards:** The rankings of all individuals in a micro-set. Often great for distributed game dynamics where you want many micro-competitions or desire to induce loyalty.

**Example:** Be the top scorers at Joe's bar this week and get a free appetizer

**Figure A5.** *Cont.*

**D9**

**Fantasy**

Giving an imaginary task to the students that requires members to assume they are in a life-threatening situation and their collaboration is needed to survive

**GM – Narrative:** Draws the players into a story within the game.

**Example:** Zombie Run, uses narrative to make the players believe that zombies are after them

**D10**

**Celebration/Reward**

A mutual reward is given for successful group work and members' efforts to achieve it

**GM – Achievement:** Segment A virtual or physical representation of having accomplished something. Often view as rewards

**Example:** A badge, a level, a reward, points

**LM – Reward:** Recognize achievement tangibly

**D10**

**Celebration/Reward**

A mutual reward is given for successful group work and members' efforts to achieve it

**GM – Fixed ratio reward schedule:** Provides rewards after a fixed number of actions. This creates cyclical nadirs of engagement

**Example:** Kill 20 ships, get a level up, get a badge, visit five locations

**LM – Amplify:** Provide learner with high output in return for little input

**D10**

**Celebration/Reward**

A mutual reward is given for successful group work and members' efforts to achieve it

**GM – Chain schedule:** Linking a reward to a series of contingencies

**Example:** Kill 10 orcs to get into the dragon's cave, every 30 min. the dragon appears

**Figure A5.** POSITIVE INTERDEPENDENCES cards.

*Appendix B.2. Templates*

The following are the templates to be used along with each category of the cards.

# CHILDREN'S PROFILE

PERSONAL DATA (See Card A1)

Number of children: _______          Ages: From ______ To _______

Number of boys: __________          Number of girls: ___________

Academic year: _______ grade

DEAFNESS-RELATED INFORMATION (See Card A2)

Degree of hearing loss:
☐ Normal hearing          ☐ Mild hearing loss
☐ Moderate hearing loss          ☐ Severe hearing loss
☐ Severe-to-profound hearing loss          ☐ Profound hearing loss

Communication:
☐ Sign language          ☐ Lip reading
☐ Oral
☐ Hearing aid          ☐ Cochlear implant

LEARNING-RELATED INFORMATION (See Card A3)

Skills/abilities: ____________________________________________
___________________________________________________________
___________________________________________________________

Learning methods and strategies:
___________________________________________________________
___________________________________________________________

Children's interests: _________________________________________
___________________________________________________________

Literacy level: _____________________________________________
___________________________________________________________

**Figure A6.** CHILDREN'S PROFILE template.

# INITIAL CONDITIONS

**TYPE OF ACTIVITY (See Card B1)**

_________________________________________________________________
_________________________________________________________________
_________________________________________________________________

**NATURE OF COLLABORATORS (See Card B2)**

☐ Peer-to-peer     ☐ Teacher student          ☐ Student-computer

**GROUP HETEROGENEITY (See Card B3)**

Size: _____ children per group

Gender:          ☐ Boys or Girls        ☐ Mixed (Boys and girls)
Academic level: ☐ Same level            ☐ Mixed (High and low level)

**SETTING OF COLLABORATION (See Card B4)**

☐ Classroom                    ☐ Home                ☐ Virtual environment
☐ Other: _________________________________________________________

**CONDITIONS OF COLLABORATION (See Card B5)**

☐ Physically                   ☐ Computer-mediated

☐ Synchronous                  ☐ Asynchronous

**PERIOD OF COLLABORATION (See Card B6)**

The activity will last _____________________________

**Figure A7.** INITIAL CONDITIONS template.

# STRUCTURE COLLABORATION

**ACTIVITIES** (See Card C1)

Main goal: _______________________________________________

_______________________________________________

Partial goals (optional): _______________________________________

_______________________________________________

_______________________________________________

Success criteria: _________________________________________

_______________________________________________

Rules: _________________________________________________

_______________________________________________

_______________________________________________

Workflow:

1. _______________________________________________

2. _______________________________________________

3. _______________________________________________

4. _______________________________________________

5. _______________________________________________

6. _______________________________________________

7. _______________________________________________

8. _______________________________________________

9. _______________________________________________

10. _______________________________________________

**Figure A8.** STRUCTURE COLLABORATION (activities) template.

# STRUCTURE COLLABORATION

**ROLES** (See Card C2)

Role 1: _______________________________________________________________________

Responsibilities: _______________________________________________________________
_______________________________________________________________________________
_______________________________________________________________________________

Role 2: ______________________________________________________

Responsibilities: _______________________________________________________________
_______________________________________________________________________________
_______________________________________________________________________________

Role 3: ______________________________________________________

Responsibilities: _______________________________________________________________
_______________________________________________________________________________
_______________________________________________________________________________

Teacher's role: ______________________________________________________

Responsibilities: _______________________________________________________________
_______________________________________________________________________________
_______________________________________________________________________________

**COMMUNICATION** (Among students and teacher) (See Card C3)

☐ Oral                    ☐ Sign language

☐ Lip-reading             ☐ Text

**Figure A9.** STRUCTURE COLLABORATION (roles and communication) template.

# STRUCTURE COLLABORATION

SHARED RESOURCES (See Card C4)

Role 1: _______________________________________________________________

Resource(s): ___________________________________________________________

_______________________________________________________________________

_______________________________________________________________________

Role 2: _______________________________________________________________

Resource(s): ___________________________________________________________

_______________________________________________________________________

_______________________________________________________________________

Role 3: _______________________________________________________________

Resource(s): ___________________________________________________________

_______________________________________________________________________

_______________________________________________________________________

**Figure A10.** STRUCTURE COLLABORATION (shared resources) template.

# POSITIVE INTERDEPENDENCES

**ROLES** (See Card D2)

Already defined when structuring collaboration.

**IDENTITY** (See Card D3)

☐ Name          ☐ Badge          ☐ Logo

☐ Motto          ☐ Flag          ☐ Image

☐ Other: _______________________________________________

**GOALS** (See Card D4)

Group goal(s): _______________________________________________

ENVIRONMENTAL (See Card D5)

Environment: _____________________________________________________

**RESOURCES** (See Card D6)

Already defined when structuring collaboration.

**Figure A11.** POSITIVE INTERDEPENDENCES (roles, identity, goals, resources) template.

# POSITIVE INTERDEPENDENCES

**TASKS** (See Card D7)

Role 1 - a. ___________________________________________________

      b. ___________________________________________________

      c. ___________________________________________________

Role 2 - a. ___________________________________________________

      b. ___________________________________________________

      c. ___________________________________________________

Role 3 - a. ___________________________________________________

      b. ___________________________________________________

      c. ___________________________________________________

**OUTSIDE ENEMY** (See Card D8)

How are groups going to compete against each other?

___________________________________________________
___________________________________________________
___________________________________________________

**FANTASY** (See Cards D9)

___________________________________________________
___________________________________________________
___________________________________________________

**CELEBRATION/REWARD** (See Card D10)

What kinds of rewards will the group get?

___________________________________________________
___________________________________________________
___________________________________________________

**Figure A12.** POSITIVE INTERDEPENDENCES (tasks, enemy, fantasy, reward) template.

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
