# Peer review of "CollabABILITY Cards: Supporting Researchers and Educators to Co-Design Computer-Supported Collaborative Learning Activities for Deaf Children"

_sustainability, doi:10.3390/su142214703_

Round 1
Reviewer 1 Report
Dear authors,
I read your submission with full passion and enjoyed reading it! Your work is valuable for helping educators/designers/developers to develop CSCL activities and tools for deaf students. Please find below my comments.
-There is no concrete research question. This makes the final message of the paper unclear: is it about proposing a set of co-design cards, developing a prototype based on them and their evaluation, or both?
-Line 71: SUS evaluation, SUS stands for what? Also, the supporting reference is missing.
-The number of headings needs revise: The background should have number 2 instead of 3. (heading number 2 is missing)
-The proposed cards, templates and process is similar to the concept of “design lenses”. A design lens is in essence a "way of viewing your design” as posed by Schell (2015) (see Jesse Schell. The art of game design: a book of lenses. CRC Press, Boca Raton, 2015. ISBN
9781466598645.) including this concept in the related work would better link and position the paper in the literature
-some demographic information about participants mentioned in sections 4.1.1 and 4.2.1 could have been provided.
-From reading sections 4.1 and 4.2, it was unclear whether each set of cards is allocated to a single learner or a group of learners. So, if we have 10 learners, do we need just 27 cards in total, or 10*27 = 270 cards? Or just a set of cards and the required information for each learner is collected in other documents?
-Section 4.2.1. would not it be more beneficial to do the evaluation of version 2.0 with the same group of participants in 4.1.1 for version 1? Further, some details on the activity used by these participants as a basis for their evaluation of the cards are missing. Did all groups use the same activity of their own selection?
-line 377: where the reader should find this activity in the rest of the paper? The section needs to be mentioned.
-line 554: the supporting reference for the Fitzgerald Key strategy is missing.
-While the evaluation results provide an image of students'/teachers’ perception of the functionality of the prototype, the evaluation of its underpinning principles such as positive interdependency, as presented in the cards, is either missing or implicit in the text. A reflection on this aspect, either in the discussion or conclusions is needed.
Good luck!
Author Response
Please see the attachment.
Thanks for your kind comments. They were all addressed.

Reviewer 2 Report
In this study, the authors use the CollabABILITY cards/templates to design activities aimed at deaf children and a particular learning goal such as literacy.
The paper is well-structured and the results are quite exciting and can be extended for children with other types of disabilities.
Minor style format corrections could be made, such as some figures not fitting quite well.
Author Response
Thanks for your review. Changes suggested by other reviewers were all addressed.
Reviewer 3 Report
- The research urgency and problem are required to be added and presented.
- Since this study is RD paradigm, the theoretical basis for the developed steps CL activity need to be mentioned and explained in more detail.
- The presentations of results and discussions need to be arranged systematically following the journal template. Thus, the discussions must be added with theoretical confirmations or dialectics to provide the validity and reliability of the results.
Author Response
Please see the attachment.
Thanks for your valuable comments.

Round 2
Reviewer 1 Report
Dear authors
I see that the provided comments on the first version of the article are addressed. Good job!
Some comments:
-first paragraph on page 5: "." should appear after [10], not before it.
-no reference for System Usability Scale (SUS) on page 7 is provided.
Regards
Author Response
Dear reviewer, thanks for your feedback. The following items show the response to your comments and observations:
Reviewer point 1: -first paragraph on page 5: "." should appear after [10], not before it.
1. The "." was deleted and added after the reference "[10]."
Reviewer point 2: -no reference for System Usability Scale (SUS) on page 7 is provided.
2. A previous reference to the System Usability Scale can be found on page 2. This is the statement: "They were used and evaluated by educators of deaf children using an adapted System Usability Scale (SUS) evaluation [13]." For this reason, no other reference was used in subsequent sections of the documents, since it was not necessary. On page 7 we talk about an adapted SUS but this adaptation was made for us for the purpose of this study, so no reference is needed.
Thanks